# Ultrasound system for precise neuromodulation of human deep brain circuits

Eleanor Martin [1,6], Morgan Roberts[1,6], Ioana F. Grigoras [2,3,4,6], Olivia Wright[1,6], Tulika Nandi [2,3,4,6], Sebastian W. Rieger [2,5], Jon Campbell [2], Tim den Boer[2,3,4], Ben T. Cox [1], Charlotte J. Stagg [2,3,4,7] ✉ & Bradley E. Treeby [1,7] ✉

We introduce an advanced transcranial ultrasound stimulation (TUS) system for precise deep brain neuromodulation, featuring a 256-element helmet-shaped transducer array (555 kHz), stereotactic positioning, individualised planning, and real-time fMRI monitoring. Experiments demonstrated selective modulation of the lateral geniculate nucleus (LGN) and connected visual cortex regions. Participants showed significantly increased visual cortex activity during concurrent TUS and visual stimulation, with high cross-individual reproducibility. A theta-burst TUS protocol produced robust neuromodulatory effects, decreasing visual cortex activity for at least 40 min post-stimulation. Control experiments confirmed these effects were specific to the targeted LGN. Our findings reveal this system's potential to non-invasively modulate deep brain circuits with unprecedented precision and specificity, offering new avenues for studying brain function and developing targeted therapies for neurological and psychiatric disorders, with transformative potential for both research and clinical applications.

Deep within the human brain are a group of grey matter structures, the basal ganglia and thalamic nuclei, which play pivotal roles in all aspects of human behaviour. Indeed, their dysregulation is pathognomonic of numerous neurological and psychiatric conditions[1]. The ability to precisely modulate neuronal activity within these areas offers potentially revolutionary therapeutic avenues for these often devastating disorders that are resistant to traditional treatments[2]. Furthermore, it unlocks insights into neural circuitry in healthy brains, presenting a plausible route to breakthrough shifts in our understanding of fundamental cognitive processes such as consciousness[3].

However, current neuromodulation techniques face significant limitations in targeting deep brain structures. Deep Brain Stimulation (DBS), though effective, is invasive and carries surgical risks[4]. Transcranial Magnetic Stimulation (TMS) and Transcranial Direct Current Stimulation (tDCS) offer non-invasive alternatives but lack the requisite depth penetration and spatial precision. TMS primarily influences cortical areas, and while its variant, Deep TMS, attempts deeper reach, it still falls short of the precision needed for specific deep brain targets[5]. tDCS, and the related technique of temporal interference, are even less focused and more diffuse, making targeted deep brain modulation a challenge[6].

[1]Department of Medical Physics and Biomedical Engineering, University College London, London, UK. [2]Nuffield Department of Clinical Neurosciences, FMRIB, Wellcome Centre for Integrative Neuroimaging, University of Oxford, Oxford, UK. [3]MRC Brain Network Dynamics Unit, Nuffield Department of Clinical Neurosciences, University of Oxford, Oxford, UK. [4]Oxford Health NHS Foundation Trust, Oxford, UK. [5]Oxford Centre for Human Brain Activity, Wellcome Centre for Integrative Neuroimaging, Department of Psychiatry, University of Oxford, Oxford, UK. [6]These authors contributed equally: Eleanor Martin, Morgan Roberts, Ioana F. Grigoras, Olivia Wright, Tulika Nandi. [7]These authors jointly supervised this work: Charlotte J. Stagg, Bradley E. Treeby. ✉e-mail: charlotte.stagg@ndcn.ox.ac.uk; b.treeby@ucl.ac.uk

Transcranial Ultrasound Stimulation (TUS) has emerged as a promising modality for non-invasive brain modulation, offering the unique advantage of deep tissue penetration[7,8]. This technique, leveraging the application of ultrasound waves, holds substantial potential for influencing neural activity in humans in both superficial and deep brain regions. However, a significant limitation of existing TUS systems, which typically employ small-aperture transducers, is the compromise in focal precision. While these systems can reach deep brain structures, the spatial resolution of the stimulation is often sub-optimal, potentially affecting a broader region than intended[9,10]. This trade-off is also linked to ultrasound frequency selection, as lower frequencies offer better skull penetration but poorer spatial resolution, while higher frequencies provide finer spatial precision but experience greater skull attenuation. This inherent trade-off between depth penetration and focal size underscores the need for advanced TUS systems capable of delivering more localised and precise neuromodulation in humans, particularly in the context of targeting deep brain structures such as the thalamus.

In addressing the limitations of focal precision in TUS, large hemispherical arrays have shown promise, particularly as evidenced in MR-guided Focused Ultrasound (MRgFUS). These arrays contain numerous transducer elements over a large aperture, allowing for finer control over the ultrasound beam with a significantly reduced focal size[11]. This technology has been successfully applied for ablative therapies, such as targeting the ventral intermediate nucleus (VIM) of the thalamus for treating essential tremor[12]. Recently, a study showed that low-power, non-thermal stimulation of the VIM and the dentato-rubro-thalamic tract (DRT) using an MRgFUS array could induce a sustained reduction of essential tremor in patients[13]. However, these arrays rely on positioning the patient using a neurosurgical frame, positioned using skull screws, and focal tissue heating for target confirmation, making them unsuitable for non-invasive, reversible neuromodulation in healthy individuals. To date, no system has been available for neuroscientific studies that can non-invasively modulate activity in the deep brain with the spatial precision required to target individual thalamic nuclei.

Here, we introduce an advanced transcranial ultrasound system that achieves high spatial precision in human deep brain neuromodulation. The system features a 256-element sparse array within an ellipsoidal helmet, enabling focal stimulation in deep brain areas. Uniquely, it is compatible with simultaneous fMRI imaging, allowing for real-time monitoring of neuromodulatory effects. A custom-designed stereotactic face and neck mask ensures precise participant positioning, while a model-based treatment planning method and an online re-planning mechanism maintain accurate targeting. This eliminates the need for a surgical frame for participant positioning and focal tissue heating for target confirmation, for the first time making a high-precision deep brain neuromodulation system available for study of the healthy brain.

We demonstrate the system's efficacy through two rigorously designed experiments in healthy human participants, targeting the lateral geniculate nucleus (LGN), one of the smallest functionally distinct nuclei of the thalamus. These experiments reveal significant and specific network effects of TUS within connected brain regions, as evidenced by changes in network activity measured using fMRI during and after task performance. Our findings underscore the potential of this advanced transcranial ultrasound system to revolutionise deep brain neuromodulation, offering new avenues for studying brain function and treating neurological and psychiatric disorders.

## Results

### Ultrasound system for precise modulation of deep brain structures

We developed an advanced transcranial ultrasound system designed for highly focal modulation of deep brain structures inside an MR scanner (Fig. 1, Fig. S2). The system is based around a semi-ellipsoidal helmet housing 256 individually-controllable transducer elements operating at a frequency of 555 kHz. A water coupling system with temperature control and hydrostatic pressure compensation ensures efficient energy transfer to the head. The helmet's dimensions and angle were optimised based on an analysis of the average adult head size to ensure comfort, accommodate a wide range of head sizes, and minimise the distance and angle of incidence to the head (Fig. S1). Numerical simulations were employed to determine the optimal element configuration within the helmet, balancing focal size and grating lobe levels while maintaining line of sight between the element positions and deep brain structures.

To characterise the system's performance, we conducted comprehensive acoustic measurements and simulations (Fig. S3). The system demonstrated the ability to steer over a wide range centred on the helmet's geometric centre, with a −3 dB focal size of 1.3 mm laterally and 3.4 mm axially at the geometric focus, giving a focal volume of 3 mm³, which is maintained across an extremely wide range of target locations (Fig. 2b, Fig. S3). Notably, this focal size is approximately 1000 times smaller than that achieved by conventional small-aperture ultrasound transducers[9,14], 30 times smaller than devices previously designed specifically for deep brain targeting in healthy humans[15], and comparable to clinical MRgFUS systems such as the ExAblate Neuro operating at similar frequencies[16]. Comprehensive compatibility testing demonstrated negligible impact of the ultrasound system on MR image quality and no influence of the MR environment on acoustic output (Fig. S4). A synchronisation setup was implemented to interleave ultrasound and MR acquisitions, effectively mitigating electromagnetic interference during simultaneous operation.

The system's small focal spot (Fig. 2) enables high-precision targeting of deep brain structures but necessitates precise alignment between the participant's head and the transducer array to ensure accurate stimulation of the desired brain region. To address this challenge, we developed a custom-designed stereotactic face and neck mask derived from individual participant MR data that can comfortably be used for healthy participants (Fig. 1, Fig. S2). The mask, fabricated using 3D printing and casting techniques, comprises two parts: a neck support and a face mask. These parts are designed to engage with specific anatomical landmarks for precise positioning: the naso-frontal angle and nasal bone anteriorly (preventing superior-inferior movement), the zygomatic bones laterally (preventing medial-lateral movement), the squamous part of the frontal bone superiorly, and the occipital bone posteriorly (preventing anterior-posterior movement)[17]. The mask is securely attached to the helmet using quick-release connectors, ensuring consistent alignment between the participant and the transducer array. This approach demonstrated high inter-session positioning repeatability, with a mean target shift of $1.50 \pm 0.70$ mm across participants and sessions, and high intra-session stability, with an average participant motion of $0.25 \pm 0.001$ mm during scans. An adjustable mirror system was also integrated into the mask, enabling participants to view a visual display unit positioned at the end of the MRI bore during experiments.

Beyond positioning, precise targeting of deep brain structures requires a treatment planning approach that accounts for the aberration and attenuation caused by the skull. We employed k-Plan, our commercially available software, to prospectively compute the driving parameters for each transducer element based on a full-wave acoustic model incorporating participant-specific skull and brain properties derived from low-dose CT scans (Fig. 2c). To maintain targeting accuracy across sessions despite small shifts in participant position, we implemented an online re-planning protocol. This protocol adjusted the pre-calculated driving phases for each element by applying geometrically calculated offsets, effectively shifting the focus to align with the desired target. Experimental validation using human skull caps

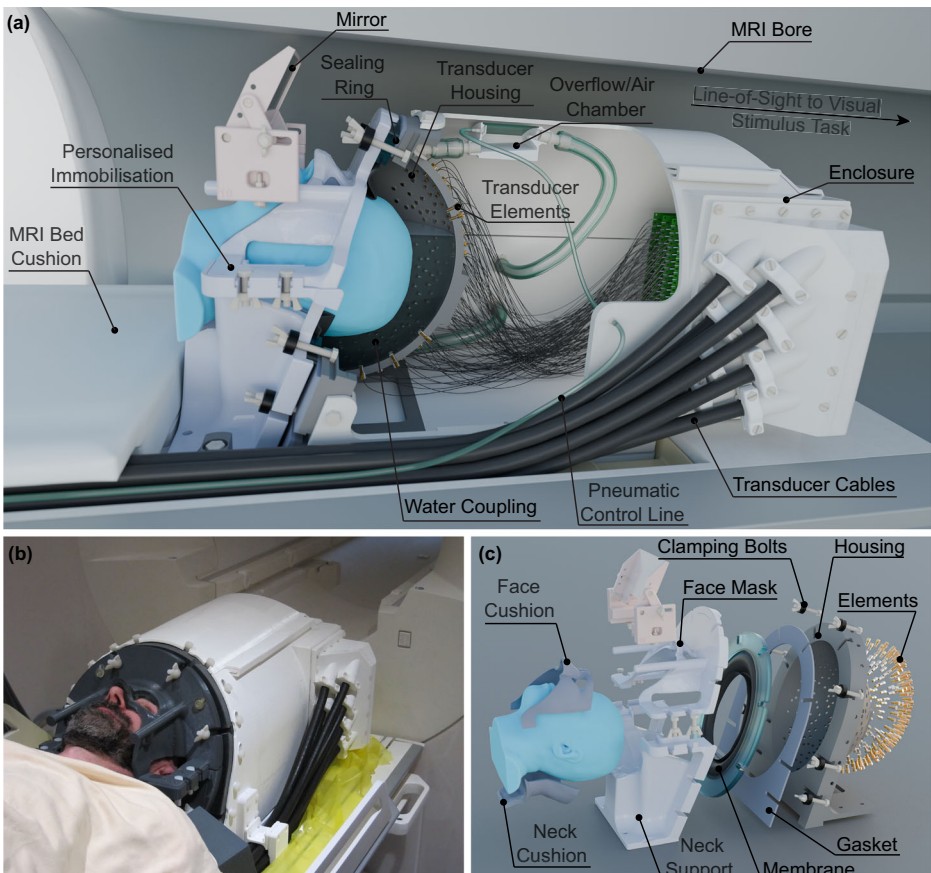

**Fig. 1 | Advanced transcranial ultrasound system setup and components.**
**a** Advanced transcranial ultrasound system within the MR bore, used for concurrent neuromodulation and functional neuroimaging. The participant is immobilised and coupled to the transducer array using water. The participant has a line of sight to a visual stimulus task outside of the bore, via a mirror. **b** Participant positioned within the system on the MR table. **c** Exploded view of the personalised immobilisation hardware, showing the sealing membrane and gaskets which retain water between the participant's head and transducer bowl.

demonstrated that the measured focal parameters were within 21% for target pressure, 0.9 mm for position, and (dx, dy, dz) = (0.2, 0.2, 0.7) mm for the −3 dB focal dimensions compared to the treatment plan (Fig. S6f). Re-planning validation showed even better agreement, confirming the validity of the isoplanatic assumption for small positional adjustments (Fig. S6g).

## Significant target engagement and prolonged network effects with TUS

To demonstrate the capabilities of our advanced transcranial ultrasound system for precise neuromodulation of deep brain structures, we targeted the well-characterised visual system. This network is an ideal testbed, as it involves both the lateral geniculate nucleus (LGN), a small, deep brain structure, and the primary visual cortex (V1), a larger, cortical region that is monosynaptically connected to the LGN (see Fig. 2d). With a volume of approximately 80 mm³, the LGN is well-suited to showcase the precise targeting capabilities of our ultrasound system, which has a −3 dB focal volume of 3 mm³ (see Fig. 2c). Simultaneously, activity in V1 is readily observable using functional magnetic resonance imaging (fMRI), providing a reliable readout of network-level effects. We employed a visual checkerboard task to activate the visual system, minimising confounds associated with participant motion or additional equipment requirements. This experimental design allowed us to demonstrate the system's ability to precisely modulate deep brain activity and observe its consequences on connected cortical regions. While our simulations predict highly focal stimulation at the LGN, we relied on observing the downstream functional effects in the connected visual cortex to demonstrate target

engagement, rather than direct measurement of activity changes within the small thalamic nucleus itself.

We conducted two experiments on seven healthy participants using a dense-sampling approach, which involves comprehensive assessments within participants to gain detailed insights into the effects of TUS. This approach was chosen to provide a rigorous understanding of both immediate (online) and lasting (offline) neuromodulatory effects, aligning with the two main types of ultrasound experiments performed in previous TUS literature.

In the first experiment, we wanted to demonstrate target engagement with TUS using an online paradigm. We, therefore, hypothesised that active TUS to the LGN would lead to significant modulation of visually-related activity in the primary visual cortex. To test this hypothesis, we employed a single-blind, pseudo-randomised, sham-controlled block design. Participants fixated on a central point while a visual checkerboard stimulus was presented on a screen. Active TUS (300 ms pulses every 3 s, target pressure of 775 kPa; see Fig. S7) was applied during half of the visual stimulation blocks, while the other half served as sham stimulation (system powered on but no ultrasound delivered), with the order of blocks pseudo-randomised to prevent order effects. These online stimulation parameters were chosen based on previous research demonstrating robust neural activation during continuous wave ultrasound application[18]. Functional MRI volumes were acquired every 3 s, interleaved with the TUS pulses. Seven participants participated in two stimulation days, with six on-target stimulation runs per day. Additionally, three off-target stimulation runs were conducted for each of the first three participants, with the TUS focus targeted at the medial dorsal nucleus (MDN), a control site close

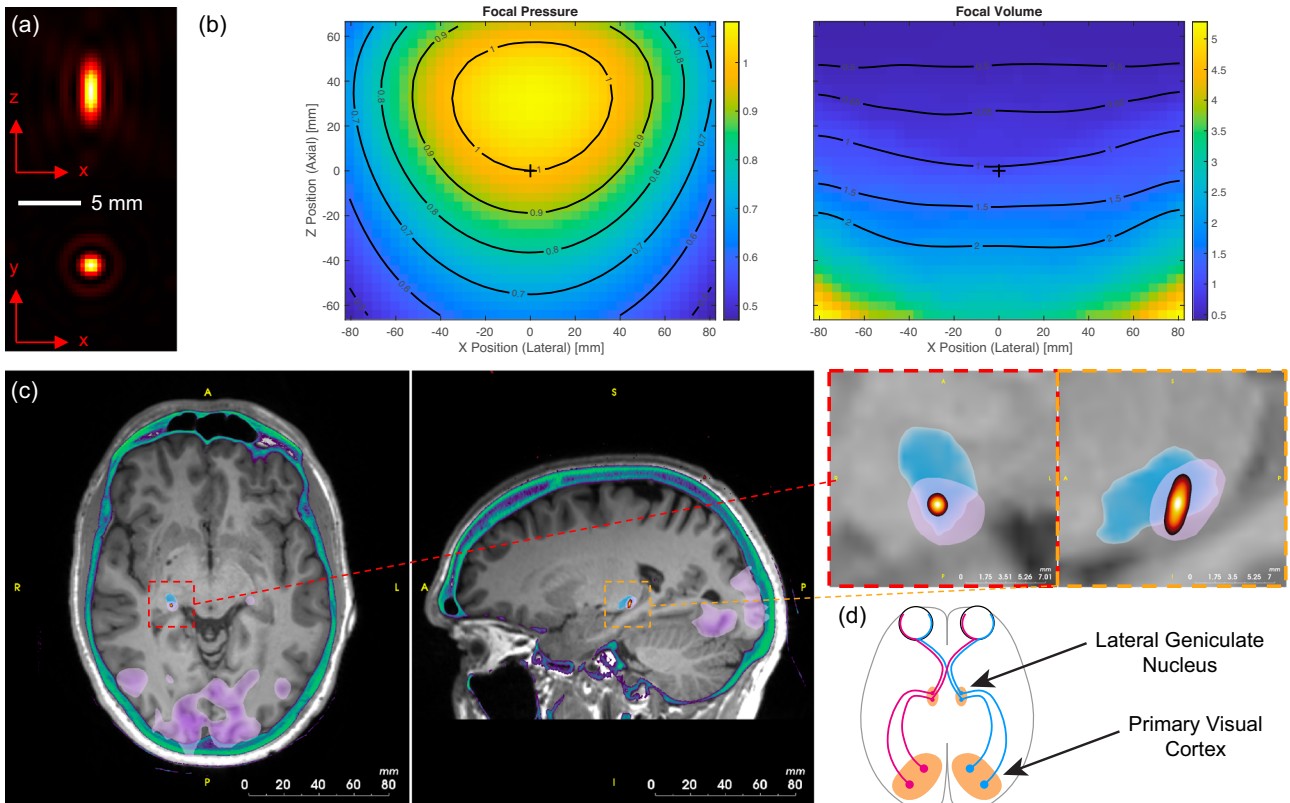

**Fig. 2 | Ultrasound focal characteristics and neuroanatomical targeting.**
**a** Simulated focal intensity for the transcranial ultrasound system in the axial (top) and lateral (bottom) planes at the geometric focus. **b** Relative focal pressure and −3 dB focal volume as a function of lateral and axial steering position. The values are normalised to the focal pressure and focal volume at the geometric centre of the helmet (position shown with a + symbol). The highly targeted focus is maintained over a very wide steering range. **c** Planning images showing the T1-weighted MR (grayscale), CT image thresholded to show the skull (green), functional activation from visual task (purple, lower-level GLM in FEAT, FSL cluster-corrected with a thresholded at $z = 3.1$, $p < 0.05$), statistical segmentation of the lateral geniculate nucleus (LGN) (blue), and −6 dB volume of the simulated target pressure (yellow/red). **d** Schematic of the visual system showing the connections between the left and right LGN and the primary visual cortices.

to the LGN (separated by an average distance of 23 mm; Fig. S9). This rigorous experimental design allowed for a robust comparison of the effects of active TUS versus sham stimulation on both the targeted LGN and its functionally connected V1.

During online stimulation, we demonstrated significantly increased task-related activity in the ipsilateral primary visual cortex during active TUS compared with sham across all participants (Fig. 3). This finding provides compelling evidence for the effective target engagement of the LGN using our advanced transcranial ultrasound system. Importantly, there were no regions of significantly altered task-related activity during LGN stimulation within the contralateral visual cortex of the task-activated network, underlining the anatomical precision of our approach and the ability to selectively modulate the targeted deep brain structure without off-target effects.

To further validate the specificity of the observed effects, for three of the participants, we conducted control experiments targeting the MDN, a thalamic nucleus close to LGN, as an active control site. TUS to this control site resulted in no significant changes in the visually-related activity within the ipsilateral occipital cortex, either on a whole-brain analysis or within the region significantly modulated by LGN stimulation, confirming that the modulation of V1 activity was indeed a consequence of precise LGN targeting and not a non-specific effect of ultrasound stimulation (Fig. 3c). The results of LGN stimulation were remarkably similar across our seven participants, both in terms of anatomical location and effect size (Fig. 3d, e). These results demonstrate the power of our advanced transcranial ultrasound system to precisely modulate deep brain activity and its

potential to elucidate the functional roles of specific neural circuits in the human brain.

In the second experiment, we aimed to establish whether TUS to the LGN could lead to long-lasting after-effects in network activity, building upon the findings of the online stimulation protocol. We employed a within-participant design in four participants with an active control site to investigate this, applying TUS using a theta-patterned approach, which has previously been shown to induce prolonged changes in cortical excitability[19–22]. This stimulation protocol, known as theta burst stimulation (TBS), consists of short, 20 ms ultrasound pulses delivered at a theta rhythm (5 Hz) for a total of 80 s. To measure brain response, participants underwent several fMRI scans during the visual checkerboard task: a baseline measurement before stimulation, an early post-stimulation task scan (scanning started 19–21 min after stimulation and lasted 16 min), and a late post-stimulation scan (scanning started approximately 140 min after stimulation and lasted 16 min). We hypothesised that TBS applied to the LGN would result in sustained modulation of visually evoked activity in the ipsilateral primary visual cortex.

Our results revealed that active TUS to the LGN led to a significant decrease in visually evoked activity in the ipsilateral primary visual cortex 40 min after stimulation compared to baseline (Fig. 4). This change in activity was no longer significant at the later post-stimulation timepoint. The anatomical location of this significantly decreased activity was very similar to the region of significant response to the online TUS, further supporting the spatial specificity of the targeted stimulation. No significant changes in

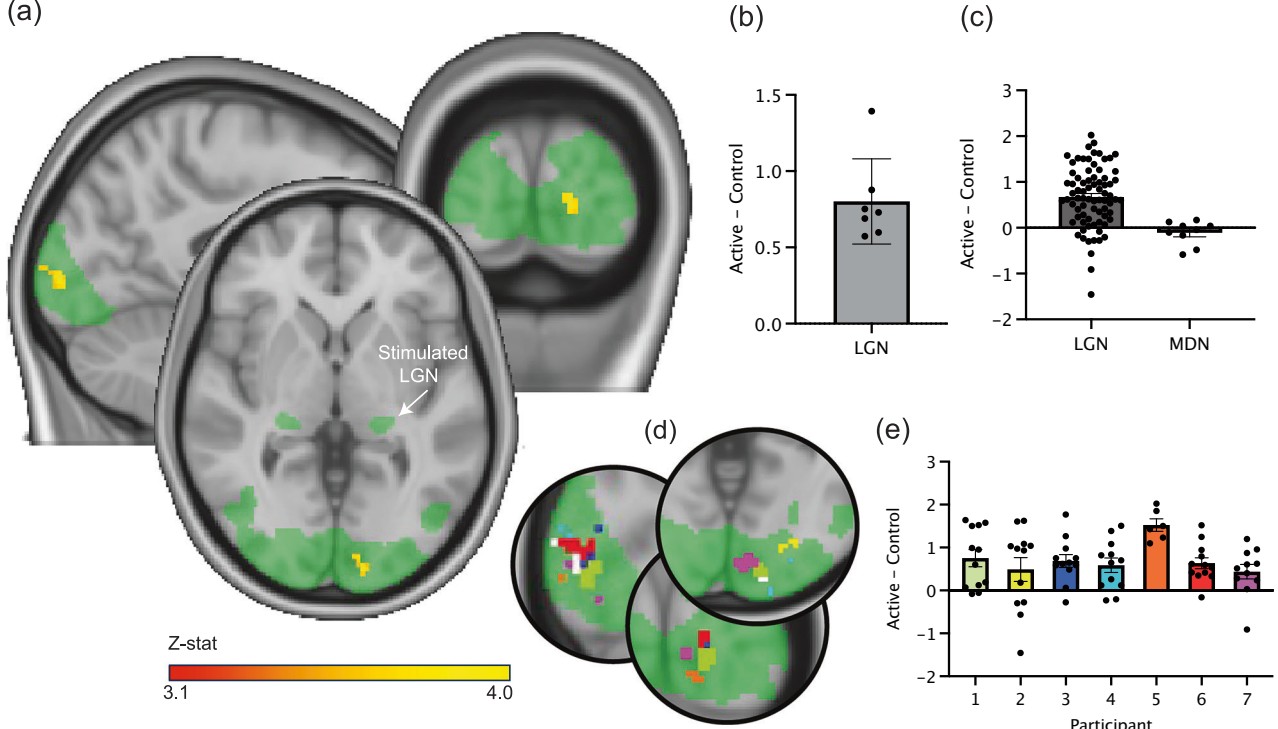

**Fig. 3 | Ultrasound stimulation to the left LGN during a visual task leads to significantly increased activity in the ipsilateral occipital cortex compared with sham. a** Group mean visually-related activation (green; z = 3.1, p < 0.05) is evident in the bilateral LGN and associated primary visual cortices during the task. Stimulation of the left (highlighted) LGN results in significantly increased task-related activity in a spatially precise region within the directly connected ipsilateral visual cortex. No changes in activity are observed in the contralateral visual cortex (FLAME 1 + 2 mixed-effects model in FEAT, FSL, cluster-corrected with a threshold of z = 3.1, p < 0.05, red-yellow). **b** Difference in task-related activity (difference in z-scored BOLD task-related change) during LGN stimulation compared with sham within the peak task-related activation region in (**a**) for each participant (n = 7). Data is represented as mean ± SD. **c** Difference in task-related activity (difference in z-scored BOLD task-related change) during LGN stimulation (left) and MDN stimulation (right) compared with sham within the peak task-related activation region for each run separately. Each data point represents a run. In total, there were 77 runs from 7 participants: 12 runs in all participants, except participant 5 (6 runs) and participant 7 (11 runs). Data is represented as mean ± SEM. **d** The peak change in task-related activity during LGN ultrasound stimulation was highly similar across individual participants. Green: mean task-related activation, white: group mean results (as in **a**). **e** Difference in task-related activity (difference in z-scored BOLD task-related change) during LGN stimulation for each run, for each participant separately. Each data point represents a run. Each participant completed 12 runs, except participant 5, who only completed 6 runs and participant 7, with 11 runs. Data is represented as mean ± SEM. Data in (**b**), (**c**) and (**e**) is extracted from the area of significant change in (**a**) and used to explain our significant findings; no statistical tests were conducted on these data.

brain activity in the primary visual cortex were observed after offline stimulation of the active control site compared to baseline, either at the early or late post-stimulation timepoints. These findings demonstrate the potential of our advanced transcranial ultrasound system to induce lasting plasticity in deep brain structures and their associated networks, opening new avenues for studying the mechanisms of neural adaptation and for developing novel therapeutic interventions.

## Discussion

In this study, we introduce a highly advanced transcranial ultrasound system that achieves unprecedented precision for deep brain neuromodulation. For the first time in humans, we demonstrate that this technology allows for specific targeting of individual thalamic nuclei non-invasively. This is evidenced by both experimental measurements and acoustic simulations confirming a small focal volume, together with the observation of spatially specific downstream effects in functionally connected regions when targeting different thalamic nuclei. Through online stimulation, we provide evidence of target engagement by demonstrating that TUS of the LGN leads to a significant increase in activity in the downstream primary visual cortex. Moreover, using a stimulation paradigm designed to induce after-effects, we observe significant modulation of activity in the ipsilateral primary visual cortex for at least 40 min following the

stimulation period. These findings represent a major step forward in our ability to precisely modulate deep brain structures and their associated networks.

The unprecedented level of spatial precision achieved by our 256-element array, along with the integration of individualised treatment planning and closed-loop targeting, marks a significant advance over prior studies. Other recent work has demonstrated notable developments in transcranial ultrasound technology, enabling electronically steered targeting of deep brain regions[15,23]. However, these proof-of-concept studies focused on modulating the subgenual cingulate cortex, without demonstrating the ability to target specific thalamic nuclei. In contrast, our simulations and experimental verifications demonstrate that our helmet transducer array achieves a focal volume nearly 30 times smaller than these systems, enabling selective targeting of structures as small as the lateral geniculate nucleus. This selectivity is reflected in the highly spatially precise changes in BOLD activity in the visual cortex, as would be expected from precise targeting of the LGN.

Another recent study demonstrated the ability to induce a sustained reduction of essential tremor in patients by targeting the VIM and the DRT using a clinical MRgFUS system[13]. While this work highlights the potential of highly focal low-power, non-thermal ultrasound stimulation for therapeutic applications, the use of a neurosurgical frame and the need for focal tissue heating to confirm the target

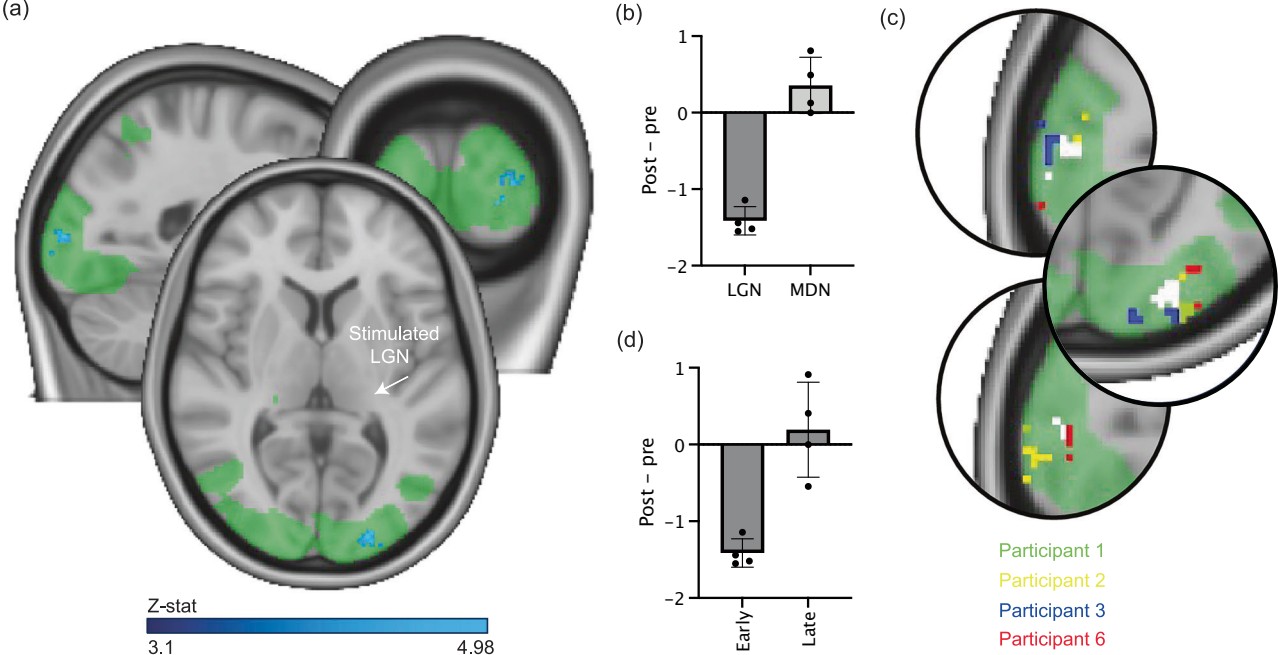

**Fig. 4 | Offline ultrasound stimulation to the left LGN significantly decreases task-related activity in the ipsilateral visual cortex during a visual checkerboard task, early after stimulation. a** Stimulation of the left (highlighted) LGN results in significantly decreased task-related activity in the directly connected ipsilateral visual cortex but not in the contralateral visual cortex (blue-light blue; FLAME 1 + 2 mixed-effects model in FEAT, FSL, cluster-corrected with a threshold of $z = 3.1$, $p < 0.05$). These significant areas overlap the mean cortical activation map of the occipital cortex during all task blocks (green; $z = 3.1$, $p < 0.05$). **b** Change in task-related activity (difference in z-scored BOLD task-related change) after LGN (left) and MDN (right) stimulation within the peak task-related activation region in (**a**) for each run separately ($n = 4$ participants; one LGN stimulation session and one MDN

stimulation session in each participant). Data is represented as mean ± SD. **c** The peak change in task-related activity during LGN ultrasound stimulation was highly similar across individual participants. Green: mean task-related activation, white: group mean results (as in **a**). **d** Change in task-related activity (difference in z-scored BOLD task-related change) after LGN stimulation within the peak task-related activation region in (**a**) for each run separately, at the early and late time-points ($n = 4$ participants; early and late timepoints of each session were compared to the baseline for each participant). Data is represented as mean ± SD. Data in (**b**) and (**d**) is extracted from the area of significant change in (**a**) from the appropriate sessions and used to explain our significant findings; no statistical tests were conducted on these data.

location limit its suitability for non-invasive neuromodulation studies in healthy participants. Our system overcomes these limitations by employing a custom-designed stereotactic face and neck mask for precise positioning and optimised model-based treatment planning and online re-planning for accurate targeting, enabling non-invasive studies in healthy populations. The advancements in spatial precision and targeting reliability open new avenues for studying the functional roles of specific deep brain nuclei and developing targeted, non-invasive therapies.

While our system has been optimised for targeting deep brain structures with high precision, it is worth considering its capabilities for other regions. The steering range of this array potentially covers a significant portion of the brain (see Fig. 2b); however, targeting superficial cortical regions presents different challenges. For such targets, many elements would have high angles of incidence to the skull (exceeding 15 degrees), which significantly reduces transmission efficiency and increases reflection. One approach to address this with the current system would be to selectively activate only elements with favourable incident angles, though this would reduce the effective aperture and increase focal size. An alternative approach for future systems would be to redistribute elements to optimise for different target locations. The optimal solution ultimately depends on the specific brain region of interest, and different array geometries and frequencies may be preferable for targeting superficial versus deep structures. Our current focus on thalamic nuclei represents an important proof of concept for high-precision deep brain targeting, which could be extended to other deep structures such as the basal ganglia, hypothalamus, and limbic regions that fall within the demonstrated steering range.

We chose to use a theta burst paradigm that has been widely used in the literature. We did not have a clear hypothesis as to whether this paradigm would cause facilitation or inhibition of the LGN. Indeed, the effects of this paradigm seem to vary across brain regions, with some studies demonstrating increased excitability and decreased inhibition when stimulating cortical regions[19,20], and others showing inhibitory after-effects[24]. The mechanisms that result in this heterogeneity are not yet clear. It is plausible that differences in stimulation intensity in the stimulated region result in different effects; other non-invasive brain stimulation paradigms show substantial non-linearity in their response profiles[25], including a reversal from inhibitory to excitatory effects with increasing intensity[26]. The significant effects of the stimulation on task-related activity were observed for at least 40 min after stimulation but were not still significant after 2 h, consistent with previous findings using the theta burst protocol[20,21].

Further research is needed to fully elucidate the mechanisms underpinning TUS and optimise stimulation parameters. A multiplicity of factors will likely influence the neuromodulatory effects of TUS, including the relative proportions of excitatory and inhibitory neurons within a target structure[27], the distribution of mechanosensitive cation channels along those neurons[28,29], and the morphology of the neuronal bodies and axons[30]. The exact contribution of these factors has yet to be established for the range of ultrasound parameters used in vivo, but as our knowledge increases, these data can be used to optimise TUS parameters based on the unique properties of deep brain structures, potentially enhancing the efficacy and specificity of the technique.

To leverage the high spatial precision, we chose to use a dense-sampling approach, averaging across large amounts of data in a relatively small number of participants, rather than across spatially varying

stimulation targets across participants. This approach has proven highly effective in finely characterising neural circuits that are difficult to study with traditional scanning methods[31–33]. However, the spatial focus of the TUS results in a very small volume of tissue within the LGN being significantly stimulated. While this spatial precision represents a step-change advance in terms of specificity for neuromodulation, the size of the focus compared with the relatively large voxel size of our fMRI sequence means that we were unlikely to be sensitive to change in activity within the LGN itself, but rather observed the effects of stimulation in directly anatomically connected structures. In addition, the lower signal-to-noise ratio for the BOLD signal in the deep brain structures means that we are less sensitive to observing activity changes in the LGN than in the relatively superficial visual cortex.

Several recent studies in animal models have demonstrated the ability of transcranial focused ultrasound stimulation to modulate activity in the visual thalamus (LGN) and produce downstream effects in the visual cortex. In sheep, reversible suppression of visually evoked potentials outlasting the stimulation period has been observed[34], while in mice, NMDAR-dependent long-term depression of thalamocortical synapses in the visual cortex has been reported[35]. Furthermore, studies in non-human primates have shown sustained effects on visual choice behaviour and gamma activity following brief LGN stimulation[36]. Our work builds upon these findings, demonstrating for the first time in humans that LGN-targeted ultrasound produces robust modulation of visual cortical activity, both during stimulation and enduring afterwards.

Our advanced TUS system enables selective and non-invasive modulation of deep brain structures in healthy adults. This approach holds promise for advancing human neuroscience beyond correlational understanding of deep structure function towards establishing causal relationships. This will allow the systematic probing of causal networks underlying the full range of cognitive processes on a participant-by-participant basis, something that had not previously been possible. TUS presents the tantalising possibility of extending our thorough understanding of the function of the deep structures in rodent models to humans. Furthermore, TUS offers the possibility of enhancing the specificity of existing therapeutic targets by elucidating the functional effects of stimulation prior to surgery, analogous to approaches that have already shown promise after surgical placement of DBS electrodes[37]. The non-invasive nature of our approach facilitates this development, as TUS can be used to modulate multiple potential therapeutic targets in patients to assess likely behavioural effects before proceeding with invasive surgery. Furthermore, TUS itself could emerge as a standalone therapeutic option for certain disorders, offering a non-invasive alternative to surgical interventions.

In conclusion, our study presents a groundbreaking advance in deep brain neuromodulation, opening new avenues for both basic research and clinical applications. However, further work is needed to fully elucidate the mechanisms of action, optimise stimulation parameters, and establish the long-term safety and efficacy of this approach. Nonetheless, the unprecedented precision and non-invasive nature of our advanced TUS system hold immense potential for revolutionising our understanding and treatment of neurological and psychiatric disorders.

## Methods

### Transcranial ultrasound system

**System overview.** The MR-compatible transcranial ultrasound system is built around 256 individual transducer elements mounted within a semi-ellipsoidal helmet (Fig. 1, Fig. S1, S2). Each element is made from an air-backed piezocomposite material with an acoustic matching layer enclosed in a custom-designed plastic housing (Sonic Concepts). These elements, each with a 3 mm aperture diameter and operating at a centre frequency of 555 kHz, are connected via 40 cm micro-coaxial cables to 8 interconnect printed circuit boards (PCBs). The selected driving frequency provides a reasonable compromise between aberration and attenuation due to the skull bone and the size of the acoustic focus[38,39]. The PCBs are connected to eight cables, each 8.2 metres long, to allow them to reach from the inside of the MR bore to the MR penetration panel. Each cable contains 32 individually shielded twisted pairs within an overall copper braid designed to mitigate electrical crosstalk and capacitive coupling. The distal ends of these cables collectively terminate in two ultrasound connectors (DL5260P, ITT Cannon) and connect to a bespoke feed-through connector mounted on the MR feed-through panel, forming an RF bond. This connector also incorporates an electrical matching network to maximise power transfer. In the MR control room, a secondary set of cables links the feed-through connector to a Verasonics Vantage 256 ultrasound drive system.

**Helmet dimensions.** The shape of the semi-ellipsoidal helmet is designed to conform to the average adult head, based on an analysis of T1-weighted MR images from 16 healthy volunteers (ages 19–42 years, 11 female). Initially, we determined a suitable inclination angle for the helmet relative to a plane perpendicular to the scanner bed. We used the approximate positions of the air-filled frontal sinus and the external occipital protuberance as reference points, taking into account the participants' head orientation in the scanner (Fig. S1a). This process yielded an optimal inclination angle of 20 degrees. We then determined the average head size by fitting an angled semi-ellipsoid to segmented head masks derived from the T1 images, resulting in average head dimensions of 206 mm in length, 157 mm in width, and 96 mm in height (Fig. S1b). To comfortably accommodate most adults while minimising the water volume and distance to the head, the helmet's interior was designed to be 40 mm larger than these average dimensions.

**Element layout.** The transducer elements were randomly distributed across the helmet surface, with each element's normal oriented towards the centre of the semi-ellipsoid. This random positioning strategy mitigates the formation of significant grating lobes, a potential concern due to the relatively low element count and the average spacing exceeding half the acoustic wavelength. To ensure line of sight to deep brain structures, we employed an offset angle of 15 degrees from the helmet's lower exterior (Fig. S1c). Additionally, we limited the element arrangement to an upper segment angle of 55 degrees, chosen based on numerical simulations to balance focal size and sidelobe height optimally (Fig. S1d). A minimum distance of 10 mm between elements was maintained for manufacturability, with additional exclusion zones to allow water connections at the highest and lowest points. The final element positions were determined from 5000 numerical simulations, selecting the configuration that minimised the relative sidelobe height (range 21%–28%).

**Helmet construction.** The ultrasound helmet was fabricated using an HP-Jet Fusion printer, employing PA12 Nylon for its high mechanical strength (Fig. S1f). Each transducer element housing was specifically designed with a flange to facilitate accurate axial placement within designated apertures in the helmet (Fig. S2a). This modular approach not only ensured precise positioning but also allowed for the individual elements to be conveniently removed and replaced if necessary, enhancing the system's maintenance and longevity. The elements were then securely coupled to the helmet using silicone adhesive. To accommodate the interconnect printed circuit boards (PCBs) and provide necessary strain relief for the cables, the helmet was integrated with a rigid enclosure (Fig. S2a). Additionally, a custom vacuum-formed liner was employed on the MRI bed, designed to prevent potential water spills from reaching the MRI bed, thereby safeguarding the integrity of the MRI system.

**Driving system.** The transcranial ultrasound system was operated using the Verasonics Vantage 256 Research Ultrasound System with an external power supply (HIFU configuration). System control and parameter adjustments were facilitated through a custom MATLAB graphical user interface (GUI). To mitigate the audibility of the ultrasound stimulation, pulse ramping was implemented using the 'states' transmit waveform type, which allows repeated transmission of a tri-state waveform with a given amplitude[40,41]. The ramp was implemented by discretising a raised cosine ramp (Tukey window) into 50 linearly spaced amplitude levels, allowing for a gradual increase or decrease in ultrasound intensity over the ramp duration. For experiments involving concurrent functional magnetic resonance imaging (fMRI), we synchronised the ultrasound system's triggering with the MR trigger out signal (Fig. S2b). This synchronisation enabled the interleaving of ultrasound stimulation with MR measurements, utilising a sparse acquisition approach to minimise interference between the two modalities.

**Subject positioning system.** For precise and repeatable positioning of participants within the helmet, we developed a custom-designed stereotactic face and neck mask, fabricated using 3D printing and casting techniques (Fig. 1, Fig. S2a). The participant's anatomical data, required for the mask's design, was obtained from T1-weighted MR images captured using a 64-channel head coil. We extracted a skin mesh from these images to derive the mask's structure. The mask comprised two parts: a neck support and a face mask, each designed to engage specific anatomical landmarks—the frontal, sphenoid, and temporal bones for the upper part, and the occipital bone and neck for the lower part—while leaving openings for the participant's eyes, ears, and mouth. Quick-release connectors facilitated the secure assembly of these parts, and a flange on the neck support was used to locate the positioning system to the helmet. Both components were 3D printed using an Ultimaker S5 printer in PLA material. To improve comfort, the surface of the neck support and the face mask were covered with a cushioning layer, cast in soft silicone rubber (Ecoflex Gel 2), providing a conformal contact with the participant. Railings on each side of the face mask were incorporated to facilitate the attachment of an adjustable mirror, enabling participants to view the visual display unit positioned at the end of the MRI bore (Fig. S2).

**Water coupling.** Acoustic coupling between the ultrasound helmet and the participant's head was achieved using deionised and degassed water, filling the space within the helmet. To facilitate effective coupling, participants' heads were shaved to prevent air bubble entrapment. The water barrier was created using a flexible silicone membrane with a central hole, mounted within an ellipse-shaped sealing flange and clamped between the participant positioning system and the helmet (Fig. 1, Fig. S2). The water was pre-heated to a physiological temperature of 37 degrees Celsius using a water conditioning unit (Sonic Concepts WCU-105) coupled to a 30 L external reservoir equipped with an additional internal heater. The helmet was filled through detachable hoses (10 metres long, 12 mm internal diameter) from its base, with an overflow chamber positioned at the top. The filling process was completed in approximately 3 min, while drainage took about 1 min. After filling, the water was not circulated. Over the course of a typical 45-min experiment, the water temperature inside the helmet was observed to decrease by approximately 4 °C.

**Air pressure control.** The hydrostatic pressure on the participant's head due to the water volume increases the contact force exerted by the positioning hardware, causing discomfort. To counteract this, we implemented a custom-built pneumatic controller to control the air pressure in the overflow chamber above the water, interconnected with the top of the helmet. This controller comprised a ported air pressure sensor with a digital readout and a diaphragm pump, both interfaced with a microprocessor. A control algorithm on the microprocessor maintained the air pressure in the chamber at a target value slightly below atmospheric pressure (set at 99,500 Pa), effectively reducing the hydrostatic pressure on the participant's head. The controller was situated in the MR control room, connected to the air chamber via a 12-metre-long tube with an internal diameter of 3 mm. This tube passed through a waveguide in the MRI penetration panel (Fig. S2).

**System calibration.** For accurate calibration of our ultrasound system's acoustic output, we followed a systematic procedure. The helmet was affixed to an automated scanning tank equipped with a five-degree-of-freedom (x, y, z, rotate, tilt) scanning arm (Fig. S6a). This arm held a calibrated 200 μm needle hydrophone (Precision Acoustics) positioned at the array's geometric centre, a location determined by the element positions and their time-of-flight measurements. We individually measured the signal from each element, extracting amplitude and phase at the driving frequency from the waveform's steady-state segment. To standardise the acoustic output across elements, we calculated amplitude scaling factors and phase offsets for each, after accounting for the hydrophone's directional response and the distance to each element. Amplitude scaling was refined through a calibration curve correlating the Verasonics apodisation parameter (which controls the on-time of the tri-state pulser) with acoustic output amplitude. Finally, we established a relationship between driving voltage and pressure using the hydrophone's calibrated sensitivity.

**Acoustic performance.** To assess the acoustic performance of our system, we conducted numerical simulations examining the acoustic output across a 16 cm steering range centred on the helmet's geometric centre (Fig. 2b). For each steering position, we evaluated the −3 dB focal size, focusing gain, and grating lobe height. A selection of these positions was experimentally validated using the same scanning setup as described in System calibration, employing a Fabry-Perot fibre optic hydrophone to minimise hydrophone directivity and spatial averaging effects. The excellent quantitative match between the simulated and experimentally measured amplitudes, profiles, and −3 dB beam sizes (Fig. S3) demonstrates the accuracy of the source definition used in numerical simulations and the near-ideal performance of the array. This close correspondence also indicates that any electrical or mechanical crosstalk is effectively negligible, thanks to the physical separation of the elements and the use of cables designed to eliminate the electrical crosstalk observed in a prototype system[42]. At the geometric focus, the −3 dB focal size was measured at 1.3 mm laterally and 3.4 mm axially. This was relatively consistent across the lateral steering range (Fig. 2b, Fig. S3b). The focal size decreased for positions inside the helmet and increased for those outside, reflecting changes in the effective aperture. Across a 5 cm steering range from the centre of the array (which covers the thalamus), the grating lobe height was less than 22%, demonstrating the effectiveness of the sparse random array design.

**MR compatibility.** To assess the impact of our ultrasound system on the MRI environment, we conducted an RF noise spectrum analysis using a Siemens 3T Prisma MRI scanner (Fig. S4a). This quality assurance scan measures the radiofrequency noise within the MRI scanner, which can be used to evaluate the system's electromagnetic compatibility. The analysis was performed under several configurations to comprehensively evaluate potential noise sources. Adding our ultrasound helmet to the MRI, either water-filled or with a participant, elevated the thermal noise floor slightly due to the increased water volume in the receive coil's field. Importantly, this did not result in any RF interference spikes, indicating the helmet's compatibility with the MRI's RF environment.

To evaluate the impact of our ultrasound helmet on MRI image quality, we conducted a second experiment comparing participant images acquired using the body coil, with and without the helmet (Fig. S4b). Analysis of these images revealed a contrast-to-noise ratio (CNR) of 30 in the reference case, compared to 26 when the helmet was used. This slight reduction in CNR indicates a modest impact of the helmet on image quality, a factor crucial for considering the helmet's integration with MRI procedures. However, when the ultrasound system was activated during image acquisition, banded artefacts were observed in the MRI images. This is attributed to the incomplete RF shielding of the ultrasound system, leading to electromagnetic interference with the MRI's signal. To mitigate this, during online Echo Planar Imaging (EPI) measurements, we employed a sparse imaging approach, interleaving MR measurements and ultrasound stimulation. This technique effectively prevented the artefacts by temporally separating the MRI acquisition from periods of ultrasound activity.

In a third experiment, we assessed the impact of the MRI environment on ultrasound transmission from the helmet (Fig. S4c). For this, we used the water-filled helmet with a blank cover, incorporating a Fabry-Perot fibre optic hydrophone positioned at its geometric centre. The helmet's driving phases were adjusted to focus on the hydrophone. Acoustic waveforms were recorded from the hydrophone under various conditions: (1) with the system on the MRI bed but outside the bore, (2) inside the MRI bore with the scanner turned off, and (3) within the bore during the acquisition of a standard EPI MRI sequence. The analysis revealed negligible differences in the acquired waveforms across these conditions, with less than 1% variation in waveform amplitude. Additionally, when the hydrophone was used to passively monitor the environment while acquiring an EPI image, no discernible signals were detected beyond the inherent noise level. This indicates that the MRI environment does not affect the acoustic output of the helmet.

**Subject positioning accuracy and repeatability.** To assess the positioning repeatability of our custom-designed stereotactic face and neck mask across sessions, we performed pairwise comparisons of positioning images from three participants, acquired over multiple sessions. Each image, including the LGN target position, was registered to a defined helmet space (see "Treatment planning" section for details). We then calculated the LGN positional differences across all images and sessions. The average shift in each Cartesian direction (in helmet coordinates) and the average overall shift (Euclidean distance) were computed. This gave average target shifts of $0.54 \pm 0.43$ mm and $0.48 \pm 0.37$ mm in the lateral directions, $1.13 \pm 0.79$ mm in the axial direction (into the helmet), and an overall average target shift of $1.50 \pm 0.70$ mm (mean ± SD). These values demonstrate a high level of positioning repeatability, comparable to, and in some cases surpassing, the precision achieved by other stereotactic positioning systems used in similar neuroimaging contexts[43,44].

To assess the efficacy of the face and neck mask in minimising head movement during MRI acquisition, participant motion was quantified through a motion correction process implemented in FEAT (FMRIB's Software Library v6.0). Functional scans underwent realignment using rigid body transformations, enabling the calculation of six movement parameters (three translations and three rotations) for each scan, relative to a designated reference volume. During representative scans (all online, on-target scans for the first three participants), the mean participant movement was $0.25 \pm 0.001$ mm (mean ± SD), demonstrating a very high level of positioning stability within sessions.

## Treatment planning
**Participant demographics.** Seven healthy participants (6 male, age range 28–54) gave their written informed consent to participate, in line with ethical approval from the East Midlands Leicester South Research

Ethics Committee (22/EM/0164). Participants had no history of neurological or psychiatric conditions and had normal or corrected-to-normal vision and were not taking any psychoactive medications. Exclusion criteria included contraindications to MRI and to non-invasive brain stimulation, including a personal or family history of seizures or epilepsy. The study was performed in accordance with the Declaration of Helsinki, except for pre-registration.

**Study visits.** The overall study was designed as two dense-sampling experiments on seven human participants split across seven visits (Fig. S5c). In visits 1–3, MR and CT planning images were obtained (see "Planning images" section). In visits 4 and 5, we conducted the online TUS experiment. In visits 6 and 7, we conducted the offline TUS experiment. Stimulation sessions were spaced by at least 1 week. Of the seven participants, four completed all seven visits, two took part in visits 1–5, and one took part in visits 1–4.

**Visual stimulus.** To elicit functional activity in the LGN and visual cortex, we utilised a radial checkerboard pattern stimulus. Stimuli were displayed on a monitor positioned at the end of the MRI bore and viewed via mirrors mounted on the coil or helmet (Fig. S2). The stimulus, set at a 50% contrast level relative to total screen luminance and reversing contrast at 7.5 Hz, was designed to balance visual engagement with comfort. This pattern has been previously shown to effectively stimulate both primary and secondary visual processing areas[45,46]. While images acquired using the head and neck coil allowed for full visibility of the stimulus, the presence of the helmet restricted the view to a rainbow-shaped section passing through the centre of the checkerboard (Fig. S4d). However, this reduced visibility still proved sufficient to robustly elicit the desired neural activity in the targeted regions (Fig. S4e). During the task, the visual stimulus was displayed in 15-s blocks followed by a 9-s break during which a fixation cross was displayed. This was repeated 20 times, giving a total task time of 8 min 9 s.

**Planning images.** For treatment planning, images were acquired across three sessions (Fig. S5a, c). In the first session, planning images were acquired using a Siemens 3T Prisma whole-body MRI scanner with a 64-channel head and neck receive coil without the ultrasound system present as follows:

- Large field-of-view (head and neck) T1-weighted MPRAGE for creation of the head and neck masks [voxel size $1 \times 1 \times 1$ mm, repetition time (TR) = 2300 ms, inversion time (TI) = 900 ms, echo time (TE) = 2.28 ms, field of view (FOV) = $192 \times 288 \times 288$ mm, sagittal, flip angle(FA) = 8°, PAT factor = 2, no fat saturation, total acquisition time (TA) = 5:58 (minutes:seconds)].
- High SNR T1-weighted MPRAGE scan optimised for grey and white matter contrast (voxel size $1 \times 1 \times 1$ mm, axial, TR = 1900 ms, TE = 3.96 ms, flip angle = 8°, TI = 912 ms, FOV = $232 \times 256 \times 192$ mm, no PAT, fat suppressed, TA = 7:21).
- Diffusion [voxel size $2 \times 2 \times 2$ mm, TR = 3600 ms, TE = 92.00 ms, FOV = $210 \times 210 \times 144$ mm, FA = 78°, TA = 6:32, MB = 3, 100 diffusion encoding directions on two shells (b = 1000 s/mm$^2$ and b = 2000 s/mm$^2$), and four b = 0 images]. This was followed by a set of three b = 0 images with reversed phase encoding direction to allow for distortion correction. Participants watched a nature video during this scan.
- Task fMRI scan acquired with a T2*-weighted 2D multiband EPI sequence (see "Visual stimulus" section) (voxel size $2.0 \times 2.0 \times 2.0$ mm, TR = 1500 ms, TE = 25 ms, FOV = $216 \times 216 \times 144$ mm, FA = 70°, TA = 8:30, MB = 3).
- B0 field map to enable distortion correction in the BOLD fMRI data (voxel size $2.5 \times 2.5 \times 2.5$ mm, TR = 590 ms, TE1 = 4.92 ms, TE2 = 7.38 ms, FOV = $210 \times 210 \times 150$ mm, FA = 46°, TA = 1:40).

The T1-weighted structural images and the single-band reference images from the T2*-weighted BOLD scans were pre-processed, bias-corrected and brain-extracted using fsl_anat and BET tools from the FMRIB Software Library (FSL)[47,48]. Fieldmaps were processed and brain-extracted using BET and fsl_prepare_fieldmap FSL tools. The task fMRI data was processed and registered to MNI standard space through first-level FEAT analysis[49]. The 4D data from each task run was modelled using a general linear model (GLM) in lower-level FEAT with one explanatory variable (EV) modelling the checkerboard presentation blocks.

In the second session, a low-dose CT scan was acquired using a GE Revolution CT scanner to obtain the participant's bony anatomy, essential for treatment planning simulations [pixel spacing = 0.45 mm (typical value), slice thickness: 0.625 mm, convolution kernel: BONEPLUS, tube current: 70 (typical value), KVP: 80]. An electron density phantom (CIRS Model 062M) was also scanned using the same acquisition and reconstruction parameters to allow a precise calibration from Hounsfield units (HU) to mass density to be determined[50].

In the third session (and during the online stimulation sessions), participants were positioned within the water-filled helmet and images were acquired using a Siemens 3T Prisma whole-body MRI scanner with the body coil. Before the scans were acquired, the shim volume was set manually to cover the brain but exclude the water in the helmet, and the GRE Brain shimming routine was manually iterated three times, followed by manual frequency adjustment. The initial manual shim was then applied to the following scans:

- T1-weighted magnetisation prepared (MPRAGE) structural scan (voxel size $1.5 \times 1.5 \times 1.5$ mm, TR = 1690 ms, TE = 3.78 ms, TI = 904 ms, FOV = $306 \times 336 \times 264$ mm, FA = 8°, TA = 5:45).
- A repeat of the MPRAGE with the inversion pulse disabled to maximise signal from the water in the ultrasound helmet for registration.
- Task fMRI scan (see "Visual stimulus" section) acquired with a sparse T2*-weighted 2D EPI sequence (voxel size $3 \times 3 \times 3$ mm, TR = 3000 ms, volume TA = 2600 ms, resulting in a 400 ms idle period in each repetition, TE = 29.0 ms, FOV = $300 \times 300 \times 114$ mm, FA = 90°, TA = 8:14).
- B0 field map to enable distortion correction in the BOLD fMRI data (voxel size $4.2 \times 4.2 \times 6$ mm, TR = 440 ms, TE1 = 4.92 ms, TE2 = 7.38 ms, FA = 45°, TA = 1:11).

**Target identification.** To determine the target voxel for the lateral geniculate nucleus (LGN), we integrated data from three sources. First, functional MRI (fMRI) data acquired during the visual task in the planning session were processed as described above, and then, the site of maximum activation within the thalamus was identified. The mean z-score for the maximum activation of the LGN across participants was $7.73 \pm 2.42$ (mean ± SD). Second, we utilised a high-resolution LGN atlas[51], which was transformed into participant space by non-linearly registering the participant's planning image with the MNI head template. Third, the thalamic nuclei were segmented using FreeSurfer, generating a parcellation of the thalamus into distinct nuclei based on a probabilistic atlas derived from histological data[52]. Based on this composite information, we manually selected the $1 \times 1 \times 1$ mm LGN voxel that exhibited the highest activation within the bilateral functional mask and overlapped with both structural LGN masks (Fig. 2c). For the active control site, our target voxel was chosen in the magnocellular medial dorsal nucleus (MDN) as identified by the FreeSurfer segmentation (Fig. S9). Targets were selected in the right hemisphere for participants 1, 4, 5 and 7, and the left hemisphere for participants 2, 3, and 6. For each participant, we selected the LGN with the most robust visually evoked activity in the planning session. The control location was then selected in the same hemisphere as the active target. All data from participants with targets in the right hemisphere were

flipped prior to group analysis, so that the stimulated LGN always appears in the left hemisphere for group analyses.

**Offline planning.** To map between helmet, brain, and CT coordinates, the acquired planning and positioning images underwent a three-step registration process (Fig. S5b). Initially, the positioning image was brain-extracted using FSL's BET tool[47]. The positioning image with the brain removed was then registered to helmet space, aligning it with a water reference image derived from CAD drawings of the helmet. Subsequently, the extracted brain images from both positioning and planning sessions were registered. Finally, the CT image, resampled to isotropic resolution using trilinear interpolation, was registered to the high-resolution planning image. This allowed the helmet position and the target positions to be mapped to the CT image coordinates for running planning simulations (Fig. 2c). All registrations were performed using FSL's FLIRT with six degrees of freedom and a mutual information cost function[53–55].

Model-based treatment planning for our ultrasound system was conducted using k-Plan, our commercially available treatment planning software. This software computes the requisite driving amplitude and phase for each transducer element to focus the acoustic array on the specified target position with the desired target pressure. It also predicts the resulting acoustic pressure and temperature fields within the head. These calculations are based on a full-wave acoustic model which accounts for the unique acoustic properties of the skull and brain tissues. A custom transducer model, matching our physical transducer's specifications, was integrated into k-Plan. Acoustic properties for each participant were mapped from low-dose CT scans, incorporating the CT calibration data. Simulations were performed with a resolution of 8 points per wavelength (Fig. S5d)[56].

The target was selected as described in Target identification, and the target acoustic pressure for all experiments and participants was set to 775 kPa. This corresponds to 20 W/cm² pulse average intensity assuming a characteristic impedance of 1.5 MRayls. In [X, Y, Z] helmet coordinates (see Fig. S2a), the mean position of the left LGN was at [−9, 23, −13] mm, the mean position of the MDN was [−5,1,2] mm, and the mean distance from the LGN to the MDN was 23 mm (Fig. S9). Across all participants and targets, the simulations predicted an in situ pressure amplitude at the target of 775 kPa, a maximum temperature increase in the brain of less than 0.2 °C, and a mechanical index $MI_{TC}$ of less than 1.9, in accordance with the ITRUSST consensus on biophysical safety[57]. The focal size in water at the average LGN target position in [X, Y, Z] helmet coordinates was: [1.4, 1.6, 1.8] mm (−3 dB free field), [2.0, 2.3, 2.6] mm (−6 dB free field), while the average focal size in situ was [1.5, 1.5, 2.2] mm (−3 dB in situ), and [2.2, 2.2, 3.1] mm (−6 dB in situ) (parameter summary included in Fig. S7)[58]. Note that the helmet axes are not necessarily aligned with the focal ellipsoid.

**Online re-planning.** To accommodate small changes in participant position relative to the helmet between sessions, we implemented a re-planning protocol, crucial for maintaining precision in targeting. Given that the ultrasound focal size, thalamic target size, and potential participant shifts are all in the 1–5 mm range, precise alignment is essential. At the beginning of each stimulation session, an additional positioning image was acquired to determine any target shift in helmet coordinates, following the same methodology as outlined in Subject positioning accuracy and repeatability (Fig. S5b). The driving phases for each helmet element, initially calculated using k-Plan, were then adjusted by adding geometrically calculated phase offsets (Fig. S5b). These adjustments aimed to shift the acoustic focus to align with the desired target position, leveraging the concept of the isoplanatic angle. This concept, borrowed from astronomy, posits that within a certain range, small shifts in participant position do not significantly alter phase distortions through the skull, thus allowing geometric adjustments to the initial phase calculations for accurate targeting[59,60].

The advantage of the geometric refocusing is that it can be computed in real time, while the k-Plan simulation must be computed offline. This approach allows us to rapidly adapt to small shifts in participant position without the need to re-run the full k-Plan simulation, while still maintaining accurate targeting. As validated in our skull experiments (Fig. S6g), the isoplanatic assumption holds well for the typical positioning differences we observed, with focal position accuracy maintained within 0.2 mm of the intended shift position. Across all participants and all online stimulation sessions, the maximum shift required was 3.0 mm, with an average value of 1.86 ± 0.56 mm.

**Experimental validation.** To validate our treatment planning workflow, we conducted experiments using four human skulls, previously sectioned along a transverse plane above the ear line to isolate the cranial portion (Fig. S6). The skulls were obtained under a material transfer agreement in accordance with the UK Human Tissue Act. Prior to each experiment, these skull caps were submerged in deionised water, degassed at −400 mbar for 48 h, and air-dried post-use. After CT scanning the skull caps, we extracted a surface mesh from the scans to design and 3D print mounts, securing the skulls in a known position relative to the helmet and the scanning tank. For each skull, we executed treatment plans targeting four positions: the geometric centre of the array and three points offset by 20 mm in each Cartesian direction within helmet coordinates. The experiments involved driving the array with an 80-cycle quasi-continuous wave signal, with phases determined by the treatment plan. Acoustic measurements were conducted in a measurement tank (described in the "System calibration" section) using a fibre optic hydrophone (FOPH), where we acquired line scans through the location of spatial peak pressure (Fig. S6d). These measurements were processed to determine the focal size, amplitude, and position, and then compared to the planned values. The results showed that, on average, the measured spatial peak pressure values were within 21% of the target pressure, and the focal position was within 0.9 mm. The mean −3 dB focal dimensions were (x, y, z) = (1.3, 1.5, 3.1) mm (Fig. S6f), with an average difference of (dx, dy, dz) = (0.2, 0.2, 0.7) mm from the planned −3 dB focal dimensions, and (dx, dy, dz) = (0.1, 0.2, 0.6) mm from the corresponding −3 dB free field focal dimensions. These findings confirm the accuracy and precision of our helmet array and the associated treatment planning software and workflow.

In an additional experiment on one skull, we validated our re-planning protocol by geometrically shifting the focus by 5 mm from two planned positions, and again acquiring line scans through the location of spatial peak pressure. The results showed that on average the measured spatial peak pressure values were within 12.5% of the unshifted values, the focal positions within 0.2 mm of the intended shift position, and the differences in −3 dB focal dimensions from the planned positions were (dx, dy, dz) = 0.03, 0.2, 0.8 mm (Fig. S6g). These findings validate the precision of our re-planning method and confirm the isoplanatic assumption for small positional adjustments.

### Online stimulation of LGN and visual cortex activity

**Experimental design.** We employed a single-blind, pseudo-randomised, sham-controlled block design to investigate the effects of transcranial ultrasound stimulation (TUS) on the visual system. During the experiment, participants fixated on a central point while a visual checkerboard stimulus was presented (see "Visual stimulus" section). Each session consisted of 20 blocks, each lasting 15 s, during which the visual stimulus was displayed (Fig. S8a). TUS was applied during 10 of these blocks, while the other 10 blocks served as sham stimulation, with the order of active and sham blocks pseudo-randomised and balanced within each session. During sham blocks, the ultrasound system remained powered on, but no ultrasound was delivered, while all other experimental conditions remained identical to the active TUS blocks. Participants were located in the MRI scanner room while the

control system was in the adjacent operator room, and they did not report being able to distinguish between active and sham conditions. The 10-ms ramp-up and ramp-down period used during active stimulation was designed to further minimise potential auditory cues. The block design was kept consistent within and across participants. Active TUS was delivered in 300 ms pulses every 3 s during active blocks (TUS was synchronised with the block timing, but not the individual checkerboard reversals), with a 10 ms ramp-up and ramp-down period to minimise auditory artefacts (see "Driving system" section). The operator of the TUS system was unblinded due to audible and visual indicators from the control system when the TUS is active. Functional MRI (fMRI) measurements were acquired every 3 s (see "Planning images" section), interleaved with the TUS pulses (Fig. S8b). Each participant underwent two online stimulation days, each including up to six MRI sessions, except for one participant who only took part in one online stimulation day. The number of stimulation runs per session varied between one and four, depending on the participant's comfort level, with a maximum of six on-target stimulation runs per day. For three participants, three additional off-target stimulation runs were conducted, with the TUS focus targeted at an active control site adjacent to the LGN (described in Target identification). This experimental design allowed for a robust comparison of the effects of active TUS versus sham stimulation on both the targeted deep brain structure (LGN) and its functionally connected cortical region (V1) while controlling for potential confounds. After each session, participants were asked to report any changes in visual perception. No changes were reported by any of the participants.

**Magnetic resonance imaging.** An identical MRI approach was used for the TUS sessions as during the third planning session (see "Planning images" section), except that more than one task sequence was performed when the participant was comfortable enough to continue with the scan.

**Image processing.** T1-weighted images were brain-extracted and bias-corrected using the BET and FAST tools from FSL[47,61]. Manual adjustments were applied to the brain extraction process when BET could not entirely eliminate water. Fieldmaps were similarly brain-extracted using BET, followed by preprocessing with the fsl_prepare_fieldmap tool. To address minor spiking artefacts which occurred on some EPI measurements when using body coil imaging (these were unrelated to the presence of the helmet, and eventually resolved by the scanner manufacturer), images were pre-processed using MELODIC independent component analysis (ICA), including motion correction using MCFLIRT, B0 field map unwarping, high-pass temporal filtering at 100 s, and no spatial smoothing[62]. Spiking artefacts were manually identified based on their spectral profiles—characterised by high power in a single volume—and removed using the fsl_regfilt tool. The resulting data were further processed through MELODIC ICA after applying a 5 mm FWHM smoothing filter. Components were then automatically labelled, and noise components were cleaned using FIX[63,64]. The cleaned data were processed and registered to the MNI standard space via first-level FEAT analysis[49]. For four participants, data were flipped after cleaning but prior to statistical analysis using the fsl_swap_dim tool to ensure that the stimulated LGN appeared in the left hemisphere across all seven participants.

**Statistical analysis.** The 4D data from each task run was modelled using a general linear model (GLM) in first-level FEAT with two explanatory variables (EVs) representing the active and sham blocks, respectively, and contrasts comparing brain activity between these blocks. Higher-level FEAT analysis was conducted using mixed effects (FLAME 1 + 2) for group analysis between participants, with separate EVs for each participant. The task mean activity map was employed as a pre-threshold mask, and a mixed-effects analysis was employed with

an automatic outlier deweighting and a cluster correction of z = 3.1 and a p threshold of 0.05. A similar analysis plan was used for the three off-target runs of three participants.

### Offline stimulation of LGN and visual cortex activity

**Experimental design.** We employed an unblinded design with an active control site to investigate the long-lasting effects of transcranial ultrasound stimulation (TUS) on the visual system (Fig. S8c). Four participants underwent two offline stimulation sessions, one targeting the lateral geniculate nucleus (LGN) and the other targeting an active control site (MDN). To measure brain response, participants underwent three functional magnetic resonance imaging (fMRI) scans: a baseline measurement before stimulation (maximum 1 h between the end of the scan and the stimulation), two early post-stimulation task scans (scanning started 19–21 min after stimulation and each scan lasted 8 min), and two late post-stimulation scans (scanning started approximately 140 min after stimulation and each scan lasted 8 min). Therefore, task fMRI data was collected on average between 20 and 40 min after stimulation for the early scan and between 140 and 160 min after stimulation for the late scan. During the stimulation session, participants fixated on a blank white screen while TUS was applied using a theta burst protocol. The stimulation lasted for a total of 80 s, with 20 ms of stimulation repeated every 200 ms (pulse repetition frequency: 5 Hz; duty cycle: 10%; 1 ms ramp-up and ramp-down). This design allowed for the assessment of both immediate and prolonged effects of TUS on visually evoked brain activity, while the active control site served to demonstrate the specificity of the stimulation effects.

**Magnetic resonance imaging.** Participants had three MRI scans (baseline, early, late) acquired using a Siemens 3T Prisma whole-body MRI scanner with a 64-channel head and neck receive coil as outlined below. The three scans were identical, except that the T1 was acquired first at baseline and last in the early and late scans.

- T1-weighted MPRAGE scan acquired in the axial plane (voxel size $1 \times 1 \times 1$ mm, TR = 1900 ms, TE = 3.96 ms, FA = 8, FOV = $232 \times 256 \times 192$ mm, TI = 912 ms, TA = 7:21).
- Resting state fMRI scan acquired with a T2*-weighted 2D multi-band EPI sequence (voxel size $2.4 \times 2.4 \times 2.4$ mm, TR = 735 ms, TE = 39.00 ms, FOV = $10 \times 210 \times 154$ mm, FA = 52°, TA = 10:00, MB = 8)[65]. Participants were asked to fixate on a white cross presented on a black screen, to blink normally, and to try not to fall asleep.
- Task fMRI scan acquired with a T2*-weighted 2D multiband EPI sequence (see "Visual stimulus" section) (voxel size $2.4 \times 2.4 \times 2.4$ mm, TR = 1500 ms, TE = 32.40 ms, FOV = $210 \times 210 \times 154$ mm, FA = 70°, TA = 8:12, MB = 4)[65]. Most MRI scans had 2 task runs, except for participant 2's late scan, which only had 1 task run (both on- and off-target).
- B0 field map to correct for distortion in the BOLD fMRI data (voxel size $2.5 \times 2.5 \times 2.5$ mm, TR = 590 ms, TE1 = 4.92 ms, TE2 = 7.38 ms, FOV = $210 \times 210 \times 150$ mm, FA = 46°, TA = 1:40).

**Image processing.** T1-weighted structural images and the single-band reference images from each T2*-weighted BOLD scan were pre-processed, bias-corrected, and brain-extracted using the fsl_anat and BET tools from FSL. Fieldmaps were processed and brain-extracted using the bet and fieldmap_prepare FSL tools. Exploratory analysis and pre-processing of the 4D task scans were performed using MELODIC ICA, which included motion correction using MCFLIRT, B0 field map unwarping, high-pass temporal filtering at 100 s, and no spatial smoothing. The FIX tool was then used to automatically label and clean the ICA components. The cleaned data was processed and registered to the MNI standard space through first-level FEAT analysis. For one participant, the data was flipped after cleaning, but before running the

statistical analysis, using the fsl_swap_dim tool. This step was performed to ensure that the stimulated LGN appeared in the left hemisphere for all four participants, facilitating group-level analyses and comparisons.

**Statistical analysis.** The 4D data from each task run was modelled using a GLM implemented in FEAT. The GLM included one EV modelling the checkerboard presentation blocks. Higher-level FEAT analyses were conducted using a mixed-effects model (FLAME 1 + 2) for repeated-measures between-participant group analysis across time, performed separately for each stimulation target (LGN and active control site). For this group analysis, we used the task mean cortical activation map as a pre-threshold mask and automatic outlier-deweighting. Significant clusters were identified using a z-threshold of 3.1 and a corrected cluster significance threshold of $p = 0.05$. This approach allowed for the identification of brain regions showing significant changes in visually evoked activity following TUS, while accounting for both within-participant and between-participant variability.

### Reporting summary

Further information on research design is available in the Nature Portfolio Reporting Summary linked to this article.

## Data availability

All data supporting the findings of this study are available within the article and its supplementary files. Any additional requests for information can be directed to and will be fulfilled by the corresponding authors. MRI sequence parameters are available from: https://doi.org/10.5281/zenodo.15360988. Unthresholded group mean statistical outputs mapped to standard space are available from: https://doi.org/10.60964/bndu-zgk7-jg52. The raw MRI data are protected and not available due to data privacy laws. Acoustic measurement data is available from: https://doi.org/10.5522/04/28687160. Source data are provided with this paper.

## Code availability

The image processing pipelines used for treatment planning and re-planning are available from: https://github.com/ucl-bug/transcranial-ultrasound-planning (https://doi.org/10.5281/zenodo.15627889). The image processing and analysis for the stimulation sessions are available from: https://git.fmrib.ox.ac.uk/grigoras/tus_mri_project (https://doi.org/10.5281/zenodo.15360988).

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

## Acknowledgements

We thank James Robertson, Adam Thomas, Silvia Schievano, Sierra Bonilla, Edward Zhang, David Marsh, Antonio Stanziola, Felix Lucka, Jiri Jaros, Marta Jaros, Nick Everdell, Kyle Morrison, and Polytimi Frangou for technical assistance. We thank the WIN radiographers, the Oxford Radiology Research Unit Team at Churchill Hospital, the Radiology Department at Great Ormond Street Hospital, Mohamed Tachrount, Aaron Hess, Howell Fu, and Will Clarke for imaging assistance. We thank Saad Jbabdi for analysis assistance. This study was supported by the Engineering and Physical Sciences Research Council, UK (EP/P008860/1, EP/P008712/1, EP/S026371/1). The k-Plan simulations were supported by the Ministry of Education, Youth and Sports of the Czech Republic through the e-INFRA CZ (ID:90254). E.M was supported by a UKRI Future Leaders Fellowship (MR/T019166/1) and in part by the Wellcome/EPSRC Centre for Interventional and Surgical Sciences (WEISS) (203145Z/16/Z). C.J.S. was supported by a Wellcome Trust Senior Research Fellowship (224430/Z/21/Z). The Wellcome Centre for Integrative Neuroimaging is supported by core funding from the Wellcome Trust (203139/Z/16/Z and 203139/A/16/Z). This study was supported by the NIHR Oxford Health Biomedical Research Centre (NIHR203316). The views expressed are those of the authors and not necessarily those of the NHS, NIHR or the Department of Health. For the purpose of open access, the author has applied a CC BY public copyright licence to any Author Accepted Manuscript version arising from this submission.

## Author contributions

E.M., M.R., O.W. and B.E.T. designed, built, and characterised the ultrasound system. T.N., C.J.S. and I.F.G. designed the study. T.N., C.J.S. and B.E.T. obtained ethics approval. I.F.G. recruited the participants. I.F.G., C.J.S., E.M., M.R., B.E.T., T.D.B. and B.T.C. acquired the data. I.F.G., C.J.S. and T.D.B. analysed the neuroimaging data. S.W.R. and J.C. optimised the imaging protocols and conducted MRI compatibility and RF shielding tests. B.E.T., C.J.S., E.M. and B.T.C. secured funding. B.E.T., C.J.S., I.F.G., E.M. and M.R. wrote the manuscript. All authors contributed to editing.

## Competing interests

B.E.T., E.M., O.W. and M.R. are authors of or have a financial interest in patent filings related to the technology described in this study. B.E.T. is a developer of the commercially available k-Plan treatment planning software used in this study, and B.E.T. and B.T.C. hold a financial interest in the software. The remaining authors declare no competing interests.
