## [Transparent Peer Review file · Nature Communications]

Ultrasound system for precise neuromodulation of human deep brain circuits

Corresponding Author: Professor Bradley Treeby

Version 0:

Reviewer comments:

Reviewer #1

(Remarks to the Author)

This is a very interesting and impactful study. It tested a TUS array system that, in theory, offers an unmatched, highly focal brain neuromodulation method. The study is well-designed, demonstrating the capability to use fMRI at off-target regions as a readout of TUS modulatory outcomes. The manuscript is well-written and easy to follow. While I commend the authors for their great efforts in conducting this study and reporting detailed TUS parameters, I have a few major and minor concerns that I hope they can address to improve the accuracy and readability of the manuscript.

Major concerns:

- (1) No detailed information is provided regarding the “sham” condition presented alongside the TUS blocks. This is crucial for readers to fully appreciate the BOLD signal changes detected during TUS.
- (2) Since BOLD signal changes at the TUS target- LGN were not quantified, the authors should clarify that the claimed high precision is based on simulations rather than actual fMRI readout at the target. One relevant piece of information would be the spatial distance between LGN and MDN nuclei, as the BOLD signal differences detected at the V1 cortex could provide insights into the spatial selectivity of the TUS modulation beam.

Minor concerns:

Results:

- (1) Figure 2. (a) It is important to emphasize that the focal size is simulated and thresholded at a particular measurement (provide such measure, FWHM?). This clarification should be included in the figure legend, especially for readers unfamiliar with the TUS field. (c) the size of fMRI activation foci depends on the statistical threshold chosen for display. This information should be provided the figure legend for reference. In Figures 3 and 4, no BOLD signal changes at LGN were reported. Wouldn't it be more informative to show the size and strength of TUS-evoked BOLD activation at the targeted LGN?
- (2) Figure 3. What is the statistical threshold of the green activation, or how are these voxels defined? This information is important for assessing the robustness of the direct modulatory effect at the target.
- (3) Figure 4. (b) and (d): Are the differences in z-scored BOLD task-related changes statistically significant? If so, which statistical test was used? Conducting statistical testing is important to support the claim: “Our results revealed that active TUS to LGN led to a significant decrease in visually-evoked” (Line 258).
- (4) Line 186: what is the sham stimulation condition? Off-target stimulation at MDN is a rigorous experimental design.
- (5) Line 287. “... achieve a focal volume nearly 30 times smaller than these systems...”. The authors should acknowledge that this statement is based on theoretical simulation rather than actual readout measurements, which, in principle, should be reflected in BOLD signal changes at different locations or targets.

Methods:

- (6) line 682-685. The manuscript stated that a resting state fMRI scan was conducted, yet no results were presented in the manuscript. Either remove this section or include the corresponding results.
- (7) Line 731-733. To help readers appreciate the robustness of the fMRI localization experiment, it is important to state the z score of the maximum activation detected within the thalamus during the planning session.
- (8) Extended data figure 4. (e). Could the authors provide the data sample size (sessions and subjects) included in generating the mean activation map?

Discussion:

(9) There is an additional layer of spatial error that could affect the precision – the alignment between T1 and fMRI BOLD images. Have the authors tested this at the LGN location across subjects when the TUS transducer is in place? This information is helpful to appreciate the influence of the presence of a transducer on fMRI image quality. This factor should be included in the discussion.

Reviewer #2

(Remarks to the Author)

This study introduces an advanced transcranial ultrasound stimulation (TUS) system designed to achieve high precision in neuromodulation of deep brain circuits. The system utilizes a 256-element helmet-shaped transducer array operating at 555 kHz, which is coupled with stereotactic positioning, individualized treatment planning, and real-time monitoring using functional MRI (fMRI). The paper demonstrates the system's ability to selectively modulate the lateral geniculate nucleus (LGN) and its connected regions in the visual cortex, leading to significant and reproducible changes in brain activity.

While the proposed helmet-shaped transducer array demonstrates impressive precision for deep brain neuromodulation, it is important to consider whether the system is capable of targeting brain regions outside of deep structures. As noted in previous studies of MRgFUS, one limitation of such systems is their difficulty in effectively targeting more superficial areas due to the constraints of focal size and penetration depth. Given that the current system is optimized for deep brain regions, it would be valuable to provide additional discussion on whether the system can be adapted or modified to target more superficial cortical regions or if any alternative strategies are being explored. Specifically, it would be beneficial to elaborate on how the system might be adjusted, in terms of transducer array configuration or frequency selection, to accommodate non-deep brain targets.

While the manuscript provides a detailed description of the system's ability to target deep brain structures, particularly the thalamus, it would be helpful to specify which other deep brain regions can be stimulated using this system within the steering range. Additionally, it would be valuable to discuss whether there are any strategies or modifications that could extend the steering range or enable stimulation of a broader range of brain regions, particularly for structures located outside this range.

In the section where the manuscript discusses the "inherent trade-off between depth penetration and focal size" (lines 51-52), it would be valuable to further elaborate on the role of fundamental frequency in this trade-off. The frequency selection in Transcranial Ultrasound Stimulation (TUS) directly influences both the depth of penetration and the spatial precision of the focal point.

In the description of the custom-designed stereotactic face and neck mask, it would be beneficial to specify the anatomical landmarks (lines 126) used in the design. While the mask is said to engage specific anatomical landmarks, providing more detail on which landmarks are used would enhance the clarity and reproducibility of the method.

The description of the pulse timing parameters for the active and sham TUS conditions (lines 183-188) provides important details, but it would be beneficial to include a visual representation of the pulse timing for both Experiment 1 and Experiment 2. A diagram illustrating the timing of the 300 ms pulses every 3 seconds alongside the fMRI data acquisition would help readers better understand the synchronization of ultrasound pulses with functional MRI scanning.

The manuscript discusses the LGN and MDN as target and control sites, respectively, but it would be helpful to include a visual representation of their spatial arrangement within the thalamus. A figure showing the relative positioning of the LGN and MDN in the anatomical context, perhaps overlaying the stimulation target regions on a structural MRI, would enhance clarity.

The manuscript describes the online and offline stimulation parameters, but further clarification on the rationale behind these choices would strengthen the study's methodological justification. Specifically, it would be helpful to clarify whether the online stimulation parameters were designed to induce an excitatory effect, as different TUS paradigms can lead to either facilitation or inhibition depending on intensity, pulse duration, and duty cycle. Additionally, the offline stimulation parameters appear to be based on prior studies using theta-burst stimulation (TBS) protocols. Given that TBS has been associated with both excitatory and inhibitory effects depending on the target and intensity, it would be valuable to discuss whether the observed neuromodulatory effects aligned with the expected outcomes based on prior literature.

Reviewer #3

(Remarks to the Author)

The authors introduce a 256 element helmet for acoustic targeting of deep brain structures. There is intense interest in this area, and novel approaches to safe and accurate targeting important to develop. From a technical perspective, this is a worthy accomplishment.

General comments:

- Very elegant experimental design with appropriate controls, revealing clear neuromodulatory effects in a well characterized

network

- Design choices for the phased array device allow for imaging data and FUS exposures to be closely interleaved, providing unique temporal resolution compared to other FUS-fMRI studies
- Validation of experimental setup and FUS device were robust
- While interesting to observe online and offline impacts on the primary visual cortex, the motivation for using two different sonication schemes is not entirely clear (beyond theta-burst TUS previously being shown to induce offline effects). Authors should expand on the motivation behind each sonication scheme.
- It's not clear from the manuscript if they actually asked subjects what they felt. Did they report any online changes in vision, visual obscurations or flashes of light? Would the device not lend itself to this type of sham controlled setup?

Specific comments:

- lines 70-71, 281: does the system described in this study have higher spatial precision than the ExAblate Neuro (670 kHz driving frequency) or other experimental hemispherical phased arrays that use MRI-guidance and contain a greater number of elements? "Unprecedented spatial precision" may not be entirely accurate; authors should provide evidence to substantiate this claim.
- lines 105-107: a more fair comparison of focal volume would be to other hemispherical (or helmet shaped) phased array devices
- lines 146-158: how tolerant are full-wave skull corrections with this device to patient positioning discrepancies? I.e. treatments can be 're-planned' to account for the patient not being positioned exactly the same as the assumed location used for skull correction computations, but in this situation corrections are not being re-computed; how much of a difference in assumed vs actual position will still lead to improved focusing?

Version 1:

Reviewer comments:

Reviewer #1

(Remarks to the Author)

The reviewers have satisfactorily addressed all of my concerns. I have one final point: the authors should consider including a description of the separately conducted target planning session and the actual sonication sessions in the Methods section. This addition is important because the presence of the TUS transducer is likely to influence the fMRI signal, such as signal drop in locations close to the transducer and some degree of functional imaging distortion, potentially at both target and off-target brain regions—a phenomenon that has been reported in preclinical studies. Such influence could potentially contribute variability in TUS outcomes, as the authors acknowledged in their responses to other questions.

For the reviewers' reference, I have pasted below their responses to the relevant question.

(9) There is an additional layer of spatial error that could affect the precision – the alignment between T1 and fMRI BOLD images. Have the authors tested this at the LGN location across subjects when the TUS transducer is in place? This information is helpful to appreciate the influence of the presence of a transducer on fMRI image quality. This factor should be included in the discussion.

We appreciate the reviewer's attention to methodological detail. We would like to clarify that all planning images (both T1 and functional MRI) used for LGN target identification were acquired using a standard head coil without the transducer present. As per standard neuroimaging practice, BOLD images were registered to T1 images for accurate targeting. The transducer was only introduced after target identification and planning had been completed. Therefore, the alignment between T1 and BOLD during planning was not affected by the presence of the transducer.

Reviewer #2

(Remarks to the Author)

I am satisfied with the response to my review comments. Thanks.

Reviewer #3

(Remarks to the Author)

The authors have addressed the points raised in the review

Response to Reviewers

We thank the Reviewers for their helpful and constructive comments. We have carefully considered the issues raised and made the appropriate changes to the manuscript, and we hope that they will all agree with us that the manuscript has been considerably strengthened as a result.

We have responded to each of the points made individually below, including any relevant excerpts from the manuscript in each case. All changes to the manuscript have been highlighted in the revised document.

Reviewer 1:

This is a very interesting and impactful study. It tested a TUS array system that, in theory, offers an unmatched, highly focal brain neuromodulation method. The study is well-designed, demonstrating the capability to use fMRI at off-target regions as a readout of TUS modulatory outcomes. The manuscript is well-written and easy to follow. While I commend the authors for their great efforts in conducting this study and reporting detailed TUS parameters, I have a few major and minor concerns that I hope they can address to improve the accuracy and readability of the manuscript.

Thank you for your positive comments about our study. We have addressed each comment below.

Major concerns:

(1) No detailed information is provided regarding the “sham” condition presented alongside the TUS blocks. This is crucial for readers to fully appreciate the BOLD signal changes detected during TUS.

We apologise for the oversight. We have now added detailed information about the sham condition to the *Methods* section under *Online Stimulation of LGN and Visual Cortex Activity: Experimental design*. We have added the following text:

“During sham blocks, the ultrasound system remained powered on but no ultrasound was delivered, while all other experimental conditions remained identical to the active TUS blocks. Participants were located in the MRI scanner room while the control system was in the adjacent operator room, and they did not report being able to distinguish between active and sham conditions. The 10 ms ramp-up and ramp-down period used during active stimulation was designed to further minimise potential auditory cues.”

(2) Since BOLD signal changes at the TUS target- LGN were not quantified, the authors should clarify that the claimed high precision is based on simulations rather than actual fMRI readout at the target. One relevant piece of information would be the spatial distance between LGN and MDN nuclei, as the BOLD signal differences detected at the V1 cortex could provide insights into the spatial selectivity of the TUS modulation beam.

The reviewer is correct that our claims of precision targeting are based on a combination of acoustic simulations, experimental measurements of focal size, and the downstream effects observed in functionally connected regions, rather than direct measurement of BOLD signal changes at the LGN itself.

To address this, we have made two additions to the main paper:

In the section *Significant target engagement and prolonged network effects with TUS*, we have added a clarifying statement at the end of the paragraph:

“While our simulations predict highly focal stimulation at the LGN, we relied on observing the downstream functional effects in the connected visual cortex to demonstrate target engagement, rather than direct measurement of activity changes within the small thalamic nucleus itself.”

In the *Discussion* section, we have revised the beginning of the first paragraph to more explicitly state how we demonstrate precision:

“This is evidenced by both experimental measurements and acoustic simulations confirming a small focal volume, together with the observation of spatially-specific downstream effects in functionally connected regions when targeting different thalamic nuclei.”

We have also added a parenthetical note specifying the average spatial distance between the LGN and MDN in the Results section where we discuss the control experiments:

“Additionally, three off-target stimulation runs were conducted for each of the first three participants, with the TUS focus targeted at the medial dorsal nucleus (MDN), a control site close to the LGN (separated by an average distance of 23 mm).”

This spatial separation information is also provided in the Methods section under *Online planning*. The differential effects observed in the visual cortex when stimulating these locations provide strong evidence for the spatial selectivity of our TUS system.

Minor concerns:

Results:

(1) Figure 2. (a) It is important to emphasize that the focal size is simulated and thresholded at a particular measurement (provide such measure, FWHM?). This clarification should be included in the figure legend, especially for readers unfamiliar with the TUS field.

Thank you. We have updated the legend for Figure 2(a) to clarify that the images show simulations. We should highlight that Figure 2(a) displays the full intensity distribution without thresholding. The figure legend now reads:

“**Simulated** focal intensity for the transcranial ultrasound system [...]”

(c) the size of fMRI activation foci depends on the statistical threshold chosen for display. This information should be provided the figure legend for reference.

We apologise for not adding this information. We have now updated the legend for Figure 2c to include that the z-score for the fMRI activation map is thresholded at $z=3.1$, $p < 0.05$ (only voxels with a z-score over 3.1 and p value less than 0.05 are displayed in the purple map in the figure). The figure legend now reads:

“(c) Planning images showing the T1-weighted MR (grayscale), CT image thresholded to show the skull (green), functional activation from visual task (purple, **thresholded at $z=3.1$, $p < 0.05$**)”

In Figures 3 and 4, no BOLD signal changes at LGN were reported. Wouldn't it be more informative to show the size and strength of TUS-evoked BOLD activation at the targeted LGN?

Thank you for this question. We did not find any significant changes in BOLD activity in the stimulated LGN in either the online experiment or the offline experiment. There might be a number of explanations for this. Firstly, the focus of the ultrasound stimulation in the LGN at -6 dB was substantially smaller than the fMRI voxel (approximately 15 mm³ versus 27 mm³ respectively). Additionally, in group mean data it is entirely possible that different voxels were stimulated across the population, further reducing our ability to detect a BOLD change at a population level. Secondly, the BOLD signal from deep brain structures has a lower signal-to-noise ratio than from superficial, cortical areas, meaning that we would be likely unable to detect a change in BOLD signal in the LGN of a similar amplitude to that observed in the visual cortex.

We have added the following to the manuscript to discuss this important point:

“While this spatial precision represents a step-change advance in terms of specificity for neuromodulation, the size of the focus compared with the relatively large voxel size of our fMRI sequence means that we were unlikely to be sensitive to change in activity within the LGN itself, but rather observed the effects of stimulation in directly anatomically connected structures. **In addition, the lower signal-to-noise ratio for the BOLD signal in the deep brain structures means that we are less sensitive to observing activity changes in the LGN than in the relatively superficial visual cortex.**”

(2) Figure 3. What is the statistical threshold of the green activation, or how are these voxels defined? This information is important for assessing the robustness of the direct modulatory effect at the target.

The green activation mask (mean activation map) shows the mean activity during the visual task (presentation of the checkerboard during both active and sham stimulation blocks) from the 12 online scans of the first 3 participants (processed as described in the paper, 36 scans in total). We used a mixed effects model in FSL (FLAME 1+2) with cluster correction at $z=3.1$ and $p=0.05$, which is a standard statistical threshold used in neuroimaging statistical analyses. This information is included in the manuscript as follows:

Methods:

“Higher-level FEAT analysis was conducted using mixed effects (FLAME 1+2) for group analysis between participants, with separate EVs for each participant. The task mean activity map was employed as a pre-threshold mask, and a mixed effects analysis was employed with an automatic outlier de-weighting and a cluster correction of $Z=3.1$ and a p threshold of 0.05.”

Figure Legend 3:

“(a) Group mean visually-related activation (green; $z=3.1$, $p<0.05$) is evident in the bilateral LGN and associated primary visual cortices during the task.”

Figure Legend 4:

“These significant areas overlap the mean cortical activation map of the occipital cortex during all task blocks (green; $z=3.1$, $p<0.05$).”

(3) Figure 4. (b) and (d): Are the differences in z-scored BOLD task-related changes statistically

significant? If so, which statistical test was used? Conducting statistical testing is important to support the claim: “Our results revealed that active TUS to LGN led to a significant decrease in visually-evoked” (Line 258).

Thank you for this question. To clarify, the areas highlighted in blue in Fig. 4a show z-scored BOLD task-related changes that are statistically significantly different at the early time point compared to baseline. To test this, we conducted a mixed-effects model in FSL FEAT (FLAME 1+2) for repeated-measures between-participant group analysis across time, performed separately for each stimulation target (LGN and active control site). For this group analysis, we used the task mean cortical activation map as a pre-threshold mask and automatic outlier-deweighting. Significant clusters were identified using a z-threshold of 3.1 and a corrected cluster significance threshold of $p=0.05$.

Figures 4b and d show the data extracted from the areas of significant effects in Figure 4a (region highlighted in blue). We are presenting them solely to help explain our significant result presented in Fig. 4a - we hope that readers will find it helpful to see the individual data points underlying the significant cluster in Fig. 4a. However, as these values are extracted directly from the statistically significant cluster in Fig. 4a, we feel that testing their significance here would be a circular analysis, and hence have not performed any statistical tests.

(4) Line 186: what is the sham stimulation condition? Off-target stimulation at MDN is a rigorous experimental design.

We have added a clarification about the sham condition at line 186:

“Active TUS [...] was applied during half of the visual stimulation blocks, while the other half served as sham stimulation (system powered on but no ultrasound delivered), with the order of blocks pseudo-randomised to prevent order effects.”

(5) Line 287. “... achieve a focal volume nearly 30 times smaller than these systems...”. The authors should acknowledge that this statement is based on theoretical simulation rather than actual readout measurements, which, in principle, should be reflected in BOLD signal changes at different locations or targets.

We have revised this line to read:

“In contrast, our simulations and experimental verifications demonstrate that our helmet transducer array achieves a focal volume nearly 30 times smaller than these systems, enabling selective targeting of structures as small as the lateral geniculate nucleus.”

Methods:

(6) line 682-685. The manuscript stated that a resting state fMRI scan was conducted, yet no results were presented in the manuscript. Either remove this section or include the corresponding results.

We have now removed this section.

(7) Line 731-733. To help readers appreciate the robustness of the fMRI localization experiment, it is important to state the z score of the maximum activation detected within the thalamus during the planning session.

We apologise for omitting this information. The mean z-score for the maximum activation of the LGN for each participant in the head coil was 7.73 (SD 2.42), showing reliable activation of the LGN during the visual checkerboard task. We have added this information to the manuscript:

“The mean z-score for the maximum activation of the LGN across participants was 7.73 \pm 2.42 (mean \pm SD).”

(8) Extended data figure 4. (e). Could the authors provide the data sample size (sessions and subjects) included in generating the mean activation map?

The data in Fig 4e top (mean activation map using the head coil) shows the mean activation map during checkerboard presentation from the planning sessions and baseline (pre-stimulation) on-target sessions from the first 3 participants (9 scans in total). The data in Fig 4e bottom (mean activation map using the body coil) shows the mean activation map during checkerboard presentation during both active and sham stimulation blocks from the first 12 online on-target scans of the first 3 participants (36 scans in total).

Discussion:

(9) There is an additional layer of spatial error that could affect the precision – the alignment between T1 and fMRI BOLD images. Have the authors tested this at the LGN location across subjects when the TUS transducer is in place? This information is helpful to appreciate the influence of the presence of a transducer on fMRI image quality. This factor should be included in the discussion.

We appreciate the reviewer's attention to methodological detail. We would like to clarify that all planning images (both T1 and functional MRI) used for LGN target identification were acquired using a standard head coil without the transducer present. As per standard neuroimaging practice, BOLD images were registered to T1 images for accurate targeting. The transducer was only introduced after target identification and planning had been completed. Therefore, the alignment between T1 and BOLD during planning was not affected by the presence of the transducer.

Reviewer 2:

This study introduces an advanced transcranial ultrasound stimulation (TUS) system designed to achieve high precision in neuromodulation of deep brain circuits. The system utilizes a 256-element helmet-shaped transducer array operating at 555 kHz, which is coupled with stereotactic positioning, individualized treatment planning, and real-time monitoring using functional MRI (fMRI). The paper demonstrates the system's ability to selectively modulate the lateral geniculate nucleus (LGN) and its connected regions in the visual cortex, leading to significant and reproducible changes in brain activity.

While the proposed helmet-shaped transducer array demonstrates impressive precision for deep brain neuromodulation, it is important to consider whether the system is capable of targeting brain regions outside of deep structures. As noted in previous studies of MRgFUS, one limitation of such systems is their difficulty in effectively targeting more superficial areas due to the constraints of focal size and penetration depth. Given that the current system is optimized for deep brain regions, it would be valuable to provide additional discussion on whether the system can be adapted or modified to target more superficial cortical regions or if any alternative strategies are being explored. Specifically, it would be beneficial to elaborate on how the system might be adjusted, in terms of transducer array configuration or frequency selection, to accommodate non-deep brain targets.

While the manuscript provides a detailed description of the system's ability to target deep brain structures, particularly the thalamus, it would be helpful to specify which other deep brain regions can be stimulated using this system within the steering range. Additionally, it would be valuable to discuss whether there are any strategies or modifications that could extend the steering range or enable stimulation of a broader range of brain regions, particularly for structures located outside this range.

Thank you for raising this important point. We have added the following paragraph to the discussion:

“While our system has been optimised for targeting deep brain structures with high precision, it is worth considering its capabilities for other regions. The steering range of this array potentially covers a significant portion of the brain (see Figure 2b); however, targeting superficial cortical regions presents different challenges. For such targets, many elements would have high angles of incidence to the skull (exceeding 15 degrees), which significantly reduces transmission efficiency and increases reflection. One approach to address this with the current system would be to selectively activate only elements with favorable incident angles, though this would reduce the effective aperture and increase focal size. An alternative approach for future systems would be to redistribute elements to optimise for different target locations. The optimal solution ultimately depends on the specific brain region of interest, and different array geometries and frequencies may be preferable for targeting superficial versus deep structures. Our current focus on thalamic nuclei represents an important proof of concept for high-precision deep brain targeting, which could be extended to other deep structures such as the basal ganglia, hypothalamus, and limbic regions that fall within the demonstrated steering range.”

In the section where the manuscript discusses the “inherent trade-off between depth penetration and focal size” (lines 51-52), it would be valuable to further elaborate on the role of fundamental frequency in this trade-off. The frequency selection in Transcranial Ultrasound Stimulation (TUS) directly influences both the depth of penetration and the spatial precision of the focal point.

Thank you - we have added the following text to elaborate on the role of the fundamental frequency:

“This trade-off is also linked to ultrasound frequency selection, as lower frequencies offer better skull penetration but poorer spatial resolution, while higher frequencies provide finer spatial precision but experience greater skull attenuation.”

In the description of the custom-designed stereotactic face and neck mask, it would be beneficial to specify the anatomical landmarks (lines 126) used in the design. While the mask is said to engage specific anatomical landmarks, providing more detail on which landmarks are used would enhance the clarity and reproducibility of the method.

We have now specified these landmarks in this description:

“These parts are designed to engage with specific anatomical landmarks for precise positioning: the nasofrontal angle and nasal bone anteriorly (preventing superior-inferior movement), the zygomatic bones laterally (preventing medial-lateral movement), the squamous part of the frontal bone superiorly, and the occipital bone posteriorly (preventing anterior-posterior movement).”

The description of the pulse timing parameters for the active and sham TUS conditions (lines 183-188) provides important details, but it would be beneficial to include a visual representation of the pulse timing for both Experiment 1 and Experiment 2. A diagram illustrating the timing of the 300 ms pulses every 3 seconds alongside the fMRI data acquisition would help readers better understand the synchronization of ultrasound pulses with functional MRI scanning.

Thank you for your suggestion. We agree that a visual representation of the pulse timings for both the online and offline experiment would help readers better understand our stimulation

and neuroimaging paradigms. We have created the figure below that illustrates the timings for the stimulation parameters in the two experiments and we have added this to the supplemental information in the paper.

Extended Data Figure 8: (a) Block structure and duration for the online stimulation experiment. The checkerboard stimulus was presented in blocks of 15 seconds separated by 9 second blocks of rest. There were 20 checkerboard blocks in total, 10 with active TUS stimulation and 10 with sham stimulation. (b) During each active TUS checkerboard presentation block, the stimulation time (300ms) was followed by fMRI data acquisition (2.6s) at a repetition time (TR) of 3 seconds. (c) Timeline of offline stimulation experiment. fMRI data during checkerboard presentation (as described above) was collected before and after the offline TUS stimulation (duration 80 seconds with 20ms TUS pulses delivered every 200ms).

The manuscript discusses the LGN and MDN as target and control sites, respectively, but it would be helpful to include a visual representation of their spatial arrangement within the thalamus. A figure showing the relative positioning of the LGN and MDN in the anatomical context, perhaps overlaying the stimulation target regions on a structural MRI, would enhance clarity.

Thank you for your suggestion. We agree that having a visual representation of the

distribution of the two thalamic nuclei will aid understanding. We have now provided a figure that illustrates the position of the left medial dorsal nucleus (yellow-orange) and the left lateral geniculate nucleus (blue) in the left thalamus (copper-grey outline, Harvard subcortical atlas) in MNI standard space. The two nuclei were segmented through the FreeSurfer pipeline described in the paper. The targeted areas are shown over the segmented nuclei: targeted MDN in green and targeted LGN in red (mean targeted area over the four participants who had both LGN and MDN stimulation). We have added this in the supplemental information.

Extended Data Figure 9: Representation of the two stimulated thalamic nuclei. The two nuclei were determined through Freesurfer segmentation (LGN in blue, MDN in yellow). The stimulated area shown here was summed up for the four participants in the offline experiment (stimulated LGN in red, stimulated MDN in green). A thalamus map from the Harvard subcortical atlas is included in copper-grey.

The manuscript describes the online and offline stimulation parameters, but further clarification on the rationale behind these choices would strengthen the study's methodological justification. Specifically, it would be helpful to clarify whether the online stimulation parameters were designed to induce an excitatory effect, as different TUS paradigms can lead to either facilitation or inhibition depending on intensity, pulse duration, and duty cycle. Additionally, the offline stimulation parameters appear to be based on prior studies using theta-burst stimulation (TBS) protocols. Given that TBS has been associated with both excitatory and inhibitory effects depending on the target and intensity, it would be valuable to discuss whether the observed neuromodulatory effects aligned with the expected outcomes based on prior literature.

Thank you for raising this important point. We chose both our online and offline stimulation parameters based on standard protocols in the literature.

For the online protocol, we selected parameters based on established research showing that continuous wave application of ultrasound effectively produces immediate neural activation during stimulation (Yoo et al., 2022).

For the offline protocol, we chose a theta burst approach that has been widely demonstrated to produce sustained after-effects in various neuromodulation paradigms (Zeng et al., 2024; Yaakub et al., 2023).

The mechanism by which TUS exerts its effect is not yet fully understood, but it seems likely that it is driven, at least in part, by modulation of ion channels. The effects of TUS therefore likely depend on a number of TUS parameters, as the reviewer suggests. The effects also likely vary from region to region due to the density of the channels in the brain region of interest, and the underlying firing rate of the region targeted. Together, this complexity means that we did not have a clear hypothesis as to the direction of the effects of our stimulation paradigms. We therefore carefully designed our visual stimuli to ensure that we would be sensitive to either an increase or a decrease in visually-evoked BOLD signal. We have added

the following to the manuscript to address this point:

“We chose to use a theta burst paradigm that has been widely used in the literature. **We did not have a clear hypothesis as to whether this paradigm would cause facilitation or inhibition of the LGN. Indeed, the** effects of this paradigm seem to vary across brain regions, with some studies demonstrating increased excitability and decreased inhibition when stimulating cortical regions, and others showing inhibitory after-effects.”

Reviewer 3:

The authors introduce a 256 element helmet for acoustic targeting of deep brain structures. There is intense interest in this area, and novel approaches to safe and accurate targeting important to develop. From a technical perspective, this is a worthy accomplishment.

General comments:

- Very elegant experimental design with appropriate controls, revealing clear neuromodulatory effects in a well characterized network
- Design choices for the phased array device allow for imaging data and FUS exposures to be closely interleaved, providing unique temporal resolution compared to other FUS-fMRI studies
- Validation of experimental setup and FUS device were robust
- While interesting to observe online and offline impacts on the primary visual cortex, the motivation for using two different sonication schemes is not entirely clear (beyond theta-burst TUS previously being shown to induce offline effects). Authors should expand on the motivation behind each sonication scheme.

Thank you for your positive comments.

As requested, we have expanded on our motivation for using different sonication schemes for online and offline effects. For the online protocol, we selected parameters based on established research showing that continuous wave application of ultrasound effectively produces immediate neural activation during stimulation (Yoo et al., 2022).

For the offline protocol, we chose a theta burst approach that has been widely demonstrated to produce sustained after-effects in various neuromodulation paradigms (Zeng et al., 2024; Yaakub et al., 2023). This approach allowed us to investigate both the immediate and prolonged effects of ultrasonic neuromodulation using paradigms optimized for each temporal domain.

We have added the following text to the section *Significant target engagement and prolonged network effects with TUS*:

“These online stimulation parameters were chosen based on previous research demonstrating robust neural activation during continuous wave ultrasound application.”

“We employed a within-participant design in four participants with an active control site to investigate this, applying TUS using a theta-patterned approach, which has previously been shown to induce prolonged changes in cortical excitability.”

- It's not clear from the manuscript if they actually asked subjects what they felt. Did they report any online changes in vision, visual obscurations or flashes of light? Would the device not lend itself to this type of sham controlled setup?

Thank you - we are pleased to have the opportunity to clarify this important point. We did indeed ask the participants if they had had any changes in visual perception, and none were reported. We have added the following to text to *Online Stimulation of LGN and Visual Cortex Activity: Experimental design*:

“After each session, participants were asked to report any changes in visual perception. No changes were reported by any of the participants.”

Specific comments:

- lines 70-71, 281: does the system described in this study have higher spatial precision than the ExAblate Neuro (670 kHz driving frequency) or other experimental hemispherical phased arrays that use MRI-guidance and contain a greater number of elements? "Unprecedented spatial precision" may not be entirely accurate; authors should provide evidence to substantiate this claim.
- lines 105-107: a more fair comparison of focal volume would be to other hemispherical (or helmet shaped) phased array devices

We agree that this is an important comparison, and have modified our description from “unprecedented precision” to “high spatial precision”. We already discuss MRgFUS systems in the manuscript (lines 95-107), but have added a more direct numerical comparison of focal volumes between our system and clinical systems like ExAblate Neuro.

“Notably, this focal size is approximately 1000 times smaller than that achieved by conventional small aperture ultrasound transducers, 30 times smaller than devices previously designed specifically for deep brain targeting in healthy humans, and comparable to clinical MRgFUS systems such as the ExAblate Neuro operating at similar frequencies.”

While our system and MRgFUS systems can achieve similar focal volumes, the key distinction remains that our system is specifically designed for non-invasive neuromodulation in healthy participants without requiring a neurosurgical frame or focal heating for targeting confirmation, as we note in the original manuscript. This represents an important advance for neuroscientific applications.

- lines 146-158: how tolerant are full-wave skull corrections with this device to patient positioning discrepancies? I.e. treatments can be 're-planned' to account for the patient not being positioned exactly the same as the assumed location used for skull correction computations, but in this situation corrections are not being re-computed; how much of a difference in assumed vs actual position will still lead to improved focusing?

We would like to clarify that our system does indeed implement a re-planning protocol based on geometric refocusing as described in the *On-line re-planning* section.

When a participant's position changes between sessions, we acquire a new positioning image and calculate the shift in target position. Rather than recomputing the full k-Plan simulation (which would be time-consuming), we use the original k-Plan simulation and then apply additional phase offsets to the driving phases based on the isoplanatic assumption. This assumption posits that within a certain range, small shifts in participant position do not significantly alter phase distortions through the skull.

We validated this approach experimentally using human skull caps, as detailed in the *Experimental validation* section and Extended Data Figure 6g. The results showed excellent agreement with the shifts up to 5 mm (the maximum tested), with focal positions maintained within 0.2 mm of the intended shift position and pressure amplitudes within 12.5% of the unshifted values.

To further clarify this point, we have added additional details in the *On-line re-planning* section:

“This approach allows us to rapidly adapt to small shifts in participant position without the need to re-run the full k-Plan simulation, while still maintaining accurate targeting. As validated in our skull experiments (Extended Data Figure 6g), the isoplanatic assumption holds well for the typical positioning differences we observed, with focal position accuracy maintained within 0.2 mm of the intended shift position.”

Response to Reviewers

The reviewers have satisfactorily addressed all of my concerns. I have one final point: the authors should consider including a description of the separately conducted target planning session and the actual sonication sessions in the Methods section. This addition is important because the presence of the TUS transducer is likely to influence the fMRI signal, such as signal drop in locations close to the transducer and some degree of functional imaging distortion, potentially at both target and off-target brain regions—a phenomenon that has been reported in preclinical studies. Such influence could potentially contribute variability in TUS outcomes, as the authors acknowledged in their responses to other questions.

For the reviewers' reference, I have pasted below their responses to the relevant question.

(9) There is an additional layer of spatial error that could affect the precision – the alignment between T1 and fMRI BOLD images. Have the authors tested this at the LGN location across subjects when the TUS transducer is in place? This information is helpful to appreciate the influence of the presence of a transducer on fMRI image quality. This factor should be included in the discussion.

We appreciate the reviewer's attention to methodological detail. We would like to clarify that all planning images (both T1 and functional MRI) used for LGN target identification were acquired using a standard head coil without the transducer present. As per standard neuroimaging practice, BOLD images were registered to T1 images for accurate targeting. The transducer was only introduced after target identification and planning had been completed. Therefore, the alignment between T1 and BOLD during planning was not affected by the presence of the transducer.

We thank the reviewer for their attention to detail. In fact, we already have described the sessions with and without the ultrasound system present in *Methods - Treatment Planning - Planning Images*. To help clarify the session without the TUS transducer, we have added the following text:

“For treatment planning, images were acquired across three sessions (Extended Data Figure 5a,c). In the first session, planning images were acquired using a Siemens 3-T Prisma whole-body MRI scanner with a 6-channel head and neck receive coil **without the ultrasound system present** as follows”

Additionally, we already directly compare and discuss the MR imaging quality with and without the ultrasound system in *Methods - Transcranial ultrasound system - MR compatibility* and *Extended Data Figure 4*.